🔓 | **Open Peer Review** | Host-Microbial Interactions | Research Article

# 16S rRNA sequencing reveals relationships among enrichment of oral microbiota in the lower respiratory tract and pulmonary nodules malignant progression

Jing Guo,[1] Jierong Han,[1] Fang Li,[1] Qiong Ma,[1] Jiawei He,[1] Fengming You,[1,2] Yifeng Ren,[1,3] Xi Fu[1,3]

**ABSTRACT**    Micro-aspiration of oral microorganisms results in considerable enrichment within the lower respiratory tract (LRT), constituting an early event in lung cancer pathogenesis. To explore the correlation between malignant risk of pulmonary nodules (PNs) and oral commensals enrichment in LRT, oral saliva and bronchial alveolar lavage fluid samples from 22 low-risk PN patients, 17 intermediate-risk PN patients, and 11 high-risk PN patients were analyzed using 16S rRNA gene sequencing. Alpha and beta diversity analyses reveal minimal variation in oral microbial diversity and abundance among patients with different risks of PN. In contrast, a significant reduction in the diversity of LRT microbiota is observed in patients at high risk of PN. Based on multi-group comparative analysis of species differences and the linear discriminant analysis effect size method, *Synergistes* and *Tannerella* were identified as the dominant bacterial genera in the oral and LRT of high-risk PN patients, respectively. The study found that the LRT microbiota of PN patients seemed to originate from the oral, and the high enrichment of oral microbiota in the lower respiratory tract was most common in high-risk PN patients. The predominant bacterial genera present in the oral cavity and LRT of patients with PN were identified through abundance variance analysis. Eight key microbial genera were found in both the oral cavity and LRT: *Streptococcus*, *Granulicatella*, *Porphyromonas*, *Bacillus*, *Neisseria*, *Alloprevotella*, *Prevotella*, and *Leptotrichia*. Notably, receiver operating characteristic analysis identified *Streptococcus*, *Granulicatella*, and *Leptotrichia* as reliable biomarkers to differentiate high-risk PN. Spearman correlation analysis confirmed that the accumulation of oral microorganisms in the LRT played an important role in the process of PN cancerization. The co-occurrence network showed that the coexistence of *Veillonella* and *Streptococcus* in the oral and LRT may be involved in the occurrence of PN, while the LRT cluster of *Rothia* occurred in high-risk PN patients. Correlation analysis among species identified microbial communities predominantly composed of *Veillonella*, which may facilitate pulmonary carcinogenesis.

**IMPORTANCE**    This study is the first to elucidate the composition and interrelationships of oral and lower respiratory tract (LRT) microbiota in patients with pulmonary nodule (PN) across varying malignancy risk levels. We conducted an analysis to investigate the correlation between the malignant potential of PNs and the enrichment of oral microbiota within the LRT. Additionally, we explored the feasibility of utilizing oral-lower respiratory commensal microbiota as biomarkers to assess the benign and malignant nature of pulmonary nodules. This study aims to provide evidence supporting early diagnosis and intervention strategies for lung cancer.

**KEYWORDS**    pulmonary nodules, oral microbiome, lower respiratory microbiome, malignant risk, biomarkers

Address correspondence to Yifeng Ren, ryftcm.dr@yahoo.com, or Xi Fu, fuxi884853@163.com.

Jing Guo and Jierong Han contributed equally to this article. Author order was determined by seniority.

The authors declare no conflict of interest.

See the funding table on p. 14.

Lung cancer (LC) continues to be the leading cause of cancer-related mortality worldwide, resulting in approximately 350 deaths per day, which is nearly 2.5 times the mortality rate of colorectal cancer, the second most common cause of cancer death (1, 2). The non-specific clinical manifestations in the early stages of LC are an important factor leading to its high mortality rate, with a 5-year relative survival rate of less than 5%, while the 5-year relative survival rate of early diagnosed LC patients can exceed 60% (3). Therefore, early identification is crucial for the early diagnosis and treatment of LC. Pulmonary nodules (PNs) are one of the main early signs of LC. With the popularization of low-dose chest computed tomography, the detection rate of PN has been increasing year by year, reaching up to 29.89% (4), accompanied by a cancer transformation rate of 5%–10% (5). However, the false positive rate is as high as 96.4%, which can lead to overdiagnosis of 0%–67% (6). Thus, creating non-invasive, highly sensitive, and specific biomarkers to accurately differentiate LC from PN, assess malignancy swiftly, and enable early intervention is crucial for improving LC patient survival. This remains a pressing clinical issue.

Prior research indicates that genetic and environmental factors primarily drive the uncontrolled proliferation of malignant tumor cells (7), while recent studies have found that lung microbiota may be involved in this process (8). Studies both domestically and internationally have shown that there were significant differences in the lower respiratory tract (LRT) microbiota characteristics among healthy individuals, PN patients, and LC patients. *Streptococcus*, *Veillonella*, and *Prevotella* were the dominant bacteria in LC patients (9). In contrast, benign PN patients' lung tissue exhibited a higher abundance of *Streptomyces*, *Moraxellaceae*, and *Stenotrophomonas*, whereas lung samples from healthy individuals were predominantly rich in *Acidocella* (10). Additionally, gram-negative bacteria such as *Haemophilus influenzae*, *Enterobacter*, and *Escherichia coli* were also enriched in LC patients (11). The incidence of small cell LC may be positively correlated with the increase of *Eubacterium* and *Clostridium* in the LRT and negatively correlated with *Pseudobutyrivibrio* (12).

The oral microbiota can disseminate to the LRT via micro-aspiration (13), exhibiting a structural similarity to the LRT microbiota (14). The oral microbiota plays a crucial role in inhibiting the colonization of respiratory pathogens on the airway mucosa, thereby preventing the further spread of pathogens within the LRT (15). Correspondingly, dysbiosis of the oral microbiota can easily lead to pathogen invasion, causing corresponding changes in the structure and abundance of the LRT microbiota, thereby increasing the risk of malignant transformation of PN (16). Compared to healthy individuals, the salivary microbiota of LC patients showed changes, particularly in the increased abundance of *Firmicutes* and *Veillonella* (17, 18), as well as the decreased abundance of *Neisseria* (19). In addition, abnormal migration of oral microbiota can lead to anomalous proliferation of disease-specific microbiota in non-dominant areas, such as LRT (20). Numerous studies have demonstrated that the abnormal migration of oral microbiota, including *Streptococcus*, *Prevotella*, and *Veillonella*, leads to their enrichment and colonization in the epithelial cells of the LRT, thereby activating key signaling pathways and disrupting the host immune phenotype, which is an early event of malignant transformation of PN into LC. However, the relationship between oral microbiota enrichment in the LRT and malignant transformation of PN has not been fully studied (21–23).

In this study, oral saliva (OS) and bronchoalveolar lavage fluid (BALF) samples were collected from patients with PN at varying levels of malignancy risk. The structural characteristics of the oral and LRT microbiota were analyzed using 16S rRNA gene sequencing, and the correlation between these microbiotas was examined. This approach allowed us to investigate the association between the enrichment of oral microbiota in the LRT and the malignancy risk of PN. Our study indicates that the enrichment of specific oral microbiota in the LRT may contribute to the malignant transformation of PN. These particular microbiota have the potential to serve as

biomarkers for the early detection of LC, thereby offering a valuable foundation for the early prevention of PN malignant transformation.

## RESULTS

### Oral microbial profiles and oral bacterial taxa associated with PN malignant risk

A cohort of 50 patients diagnosed with PN and admitted to Chengdu Integrated Traditional Chinese and Western Medicine Hospital in Sichuan Province between January 2023 and September 2023 were selected for this study. Following a comprehensive risk assessment, the patients were categorized into three groups: 22 patients in the low-risk group, 17 in the medium-risk group, and 11 in the high-risk group. BALF and OS samples were collected from each patient across the three risk groups (Table S1). The microbial composition of these samples was subsequently analyzed using 16S rRNA gene sequencing, yielding a total of 20,772 amplicon sequence variants (ASVs). The three groups of BALF samples had a total of 8,116 ASVs, including 4,129 in the low-risk group (BALFL), 2,322 in the medium-risk group (BALFM), and 1,665 in the high-risk group (BALFH). The three groups of OS samples had a total of 12,656 ASVs, of which 5,438 were in the low-risk group (OSL), 4,547 were in the medium-risk group (OSM), and 2,671 were in the high-risk group (OSH).

*Firmicutes*, *Bacteroidota*, *Actinobacteriota*, *Proteobacteria*, and *Fusobacteriota* were the most common five phyla of oral microbiota (Fig. 1A). At the genus level of microbial composition, the top nine relative abundances in the OS group were *Streptococcus*, *Prevotella*, *Veillonella*, *Rothia*, *Actinomyces*, *Porphyromonas*, *Neisseria*, *Granulicatella*, and *Haemophilus* (Fig. 1B; Table 1).

Alpha diversity analysis was conducted on the oral microbiota of different risk stratification for PN, and a difference test was conducted among three groups on the obtained indexes. It was found that the Chao1 ($P = 0.6708$), Sobs ($P = 0.6658$), pd ($P = 0.5021$), Simpson ($P = 0.172$), and Shannon ($P = 0.3352$) index were not statistically significant (Fig. S1). An Adonis analysis, which is a nonparametric multivariate test, was conducted using UniFrac distances. As revealed by the principal coordinate analysis (PCoA) of Beta diversity, the contribution rate of PCoA1 was 17.7%, and the contribution rate of PCoA2 was 9.83% ($R^2 = 0.0247$, $P = 0.985$), indicating that the clustering of visual evaluation samples was relatively scattered when the unweighted analysis was performed. Weighted analysis was performed and indicated that the contribution rates of PCoA1 and PCoA2 were 29.10% and 12.91%, respectively, and there was no obvious

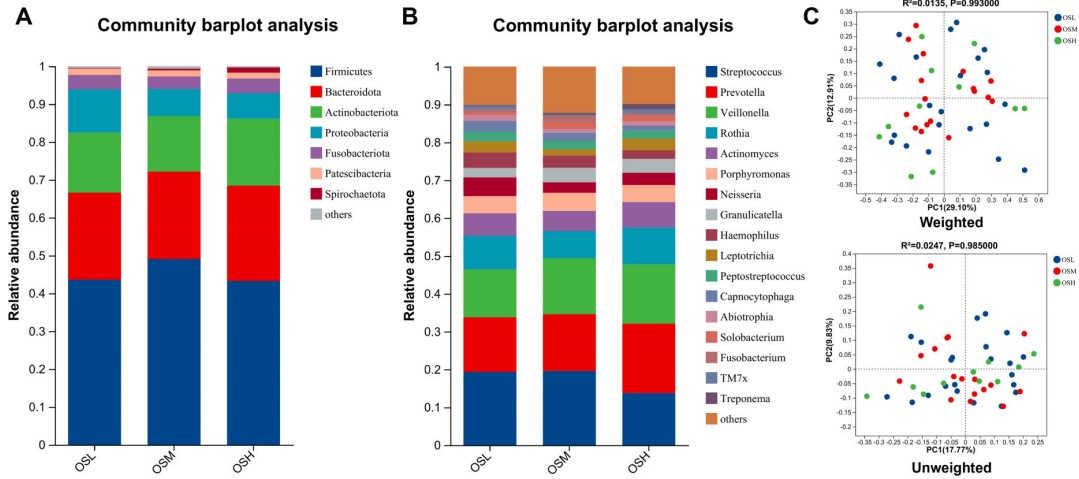

**FIG 1** Structural characteristics and comparative analysis of oral microbiota in low-, medium-, and high-risk PN. (A) Proportions of bacterial phylum levels. (B) Proportions of bacterial genera levels. (C) Beta diversity differences estimated by principal coordinate analysis (PCoA). Top, weighted PCoA plots; bottom, unweighted PCoA plots. OSL group (blue dots); OSM group (red dots); OSH group (green dots); each dot represents a single sample.

**TABLE 1** The proportions of bacterial phylum levels and genera levels in OS[a]

| Category | Phylum/genera | OSL (%) | OSM (%) | OSH (%) |
|---|---|---|---|---|
| Bacterial phylum | *Firmicutes* | 43.6 | 49.1 | 43.3 |
| | *Bacteroidota* | 23.0 | 23.1 | 25.1 |
| | *Actinobacteriota* | 15.9 | 14.7 | 17.8 |
| | *Proteobacteria* | 11.5 | 7.2 | 6.7 |
| | *Fusobacteriota* | 3.8 | 3.2 | 3.9 |
| Bacterial genera | *Streptococcus* | 19.3 | 19.6 | 13.8 |
| | *Prevotella* | 14.4 | 15.0 | 18.3 |
| | *Veillonella* | 12.7 | 14.8 | 15.7 |
| | *Rothia* | 8.9 | 7.1 | 9.7 |
| | *Actinomyces* | 5.9 | 5.3 | 6.7 |
| | *Porphyromonas* | 4.5 | 4.8 | 4.5 |
| | *Neisseria* | 4.6 | 2.8 | 3.2 |
| | *Granulicatella* | 2.5 | 3.8 | 3.7 |
| | *Haemophilus* | 4.1 | 3.2 | 2.3 |

[a]OS: oral saliva; OSL: low-risk group of OS; OSM: medium-risk group of OS; OSH: high-risk group of OS.

separation of the bacterial communities among the three groups ($R^2 = 0.0135$, $P = 0.993$). Additionally, no statistically significant differences in microbial species were observed among the three groups (Fig. 1C).

The Kruskal-Wallis H rank sum test was used to differentiate between taxa in the various groups at different classification levels. Dunn's test was used for comparison between the two groups. Among the five genera that exhibited differences across the three groups(Fig. 2A), the Dunn's test revealed a significant increase in the genus *Synergistes* in patients classified as high-risk for PN ($P < 0.01$). Additionally, a significant difference was observed when comparing this group to the medium-risk PN population ($P < 0.01$; Fig. 2B; Fig. S2).

## LRT microbial profiles and LRT bacterial taxa associated with PN malignant risk

The relative abundance of five phyla—*Proteobacteria*, *Firmicutes*, *Bacteroidota*, *Actinobacteriota*, and *Fusobacteriota* was notably high in the LRT microbiota (Fig. 3A). Additionally, 10 genera, including *Achromobacter*, *Streptococcus*, *Pedobacter*, *Veillonella*, *Prevotella*, *Undibacterium*, *Pseudomonas*, *Porphyromonas*, *Haemophilus*, and *Neisseria*, exhibited relatively high abundance in the LRT microbiota (Fig. 3B; Table 2).

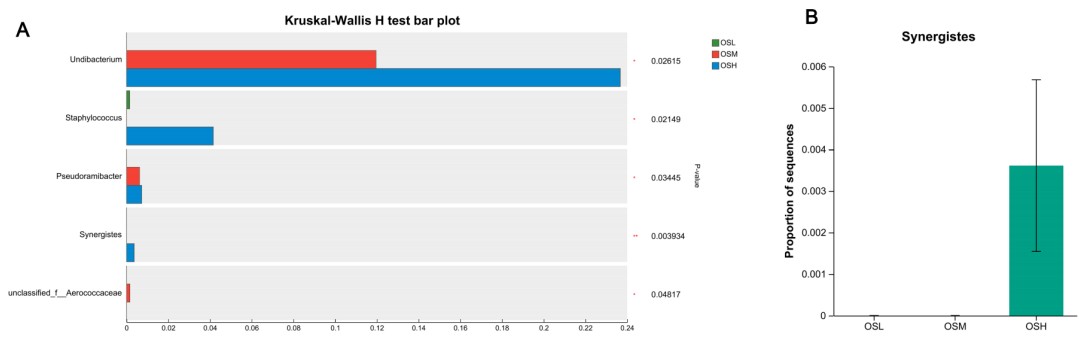

**FIG 2** Comparative analysis of oral microbiota in low-, medium-, and high-risk PN. (A) Comparative analysis of species differences in three groups. (B) Differential oral bacteria genus.

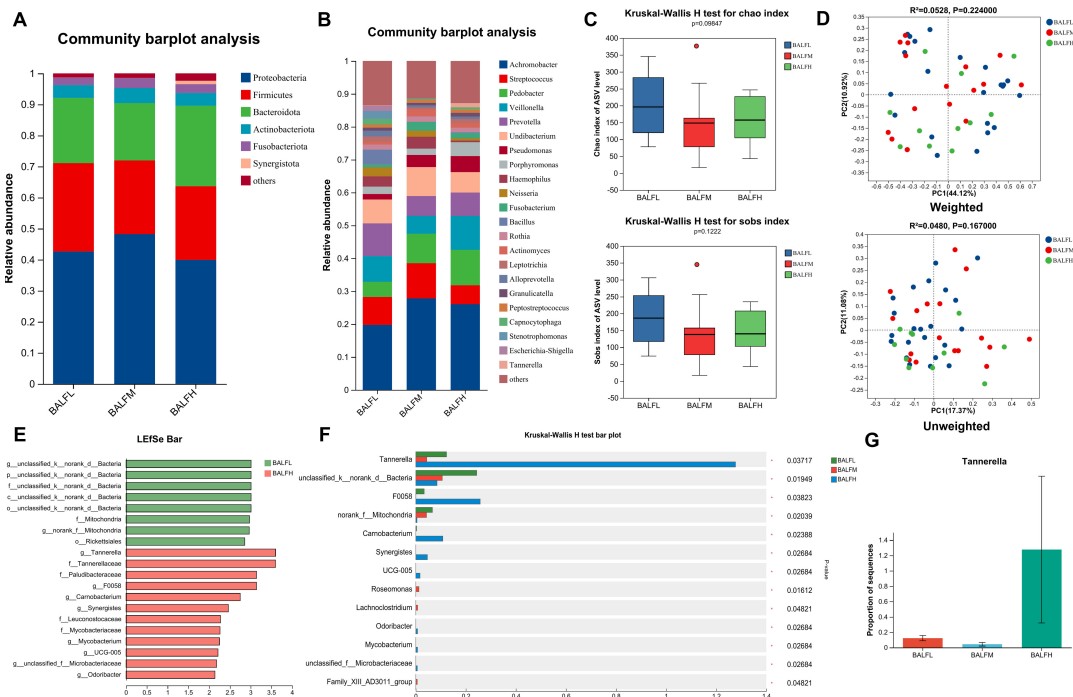

**FIG 3** Structural characteristics and comparative analysis of LRT microbiota in low-, medium-, and high-risk PN. (A) Proportions of bacterial phylum levels. (B) Proportions of bacterial genera levels. (C) Differences in alpha diversity for chao1, Sobs index. (D) Beta diversity differences estimated by PCoA. Top, weighted PCoA plots; bottom, unweighted PCoA plots. BALFL group (blue dots); BALFM group (red dots); BALFH group (green dots); each dot represents a single sample. (E) Linear discriminant analysis (LDA > 3). (F) Comparative analysis of species differences in three groups. (G) Differential oral bacteria genus.

## High-risk group of BALF

As indicated by the Chao1 and SOBS indices, the bacterial community in LRT of medium-risk and high-risk PN patients were lower than low-risk PN patients ($P = 0.0985$ and $P = 0.1222$ for Chao1 and SOBS, respectively; Fig. 3C). The clustering of samples, as assessed visually in the unweighted analysis, was more dispersed than in the weighted analysis; there was no significant difference in the LRT microbiota species composition among the three groups ($R^2 = 0.0480$, $P = 0.1670$ for unweighted analysis; $R^2 = 0.0528$, $P = 0.2240$ for weighted analysis; Fig. 3D). The linear discriminant analysis (LDA) effect size (LEfSe) method and intergroup difference testing method were utilized. Thirteen genera exhibited significant differences among the three groups (Fig. 3E and F). However, Dunn's test revealed that *Tannerella* was significantly elevated in high-risk patients with PN ($P < 0.01$), and there was also a significant difference when compared to the moderate-risk PN population ($P < 0.01$; Fig. 3G; Fig. S3)

## Enrichment of oral microbiota in the LRT indicated the potential of microbiome biomarkers for distinguishing PN malignant risk

To verify the relationship between oral microbiota and LRT microbiota, Venn analysis was conducted on the ASVs of oral and LRT microbiota in individuals with low-, medium-, and high-risk PN. The analysis revealed that 159 ASVs (37.9%) of the LRT microbiota in low-risk PN patients overlapped with their oral microbiota. In patients with moderate risk for PN, 127 ASVs (34.8%) of the LRT microbiota were shared with their oral microbiota. For high-risk PN patients, 117 ASVs (49.6%) of the LRT microbiota overlapped with their oral microbiota (Fig. 4A). The findings suggest a significant overlap between oral microorganisms and those present in the LRT among patients with low, medium, and high risk of PN. Notably, the overlap rate approached 50% in the high-risk

**TABLE 2** The proportions of bacterial phyla and genera levels in LRT[a]

| Category | Phylum/genera | BALFL (%) | BALFM (%) | BALFH (%) |
|---|---|---|---|---|
| Bacterial phylum | *Proteobacteria* | 42.6 | 48.2 | 39.9 |
| | *Firmicutes* | 28.5 | 23.7 | 23.7 |
| | *Bacteroidota* | 21.0 | 18.5 | 25.9 |
| | *Actinobacteriota* | 4.1 | 4.8 | 4.1 |
| | *Fusobacteriota* | 2.4 | 3.2 | 2.8 |
| Bacterial genera | *Achromobacter* | 19.8 | 27.8 | 26.0 |
| | *Streptococcus* | 8.5 | 10.7 | 5.8 |
| | *Pedobacter* | 4.6 | 9.0 | 10.8 |
| | *Veillonella* | 7.8 | 5.4 | 10.3 |
| | *Prevotella* | 10.0 | 6.0 | 7.1 |
| | *Undibacterium* | 7.2 | 8.8 | 6.2 |
| | *Pseudomonas* | 1.7 | 3.7 | 4.9 |
| | *Porphyromonas* | 2.2 | 1.9 | 4.2 |
| | *Haemophilus* | 3.1 | 3.6 | 0.5 |
| | *Neisseria* | 2.7 | 1.9 | 0.7 |

[a]BALFL: low-risk group of BALF; BALFM: medium-risk group of BALF; BALFH: high-risk group of BALF.

group, potentially attributable to the translocation and subsequent enrichment of oral microflora within the LRT.

Analyzing the α diversity of oral and LRT microbiota in the population with PN, it was found that there was a significant difference in the Chao1, Shannon, and Shannon indexes between the oral and LRT microbiota of patients with moderate- and high-risk PN ($P < 0.01$ and $P = 0.0455$ for moderate and high risk, respectively; Fig. 4B). Notably, there was an increasing trend in the difference of the Chao1 index between the oral and LRT microbiota. A comparative analysis of β diversity in the oral and LRT microbiota revealed that, following weighted analysis, the microbiota of patients with medium- and high-risk PN exhibited closer clustering (Fig. 4C). This suggests that the structural composition of the oral and LRT microbiota in these patients is more similar.

The predominant bacterial genera present in the oral cavity and LRT of patients with PN were identified through abundance variance analysis. Eight key microbial genera were found in both the oral cavity and LRT: *Streptococcus*, *Granulicatella*, *Porphyromonas*, *Bacillus*, *Neisseria*, *Alloprevotella*, *Prevotella*, and *Leptotrichia*. Among these, *Streptococcus*, *Granulicatella*, and *Porphyromonas* exhibited no significant differences in abundance between the oral cavity and LRT, suggesting that these three bacterial genera are enriched in both anatomical sites (Table S2).

Using the above eight significantly different bacterial genera for receiver operating characteristic (ROC) diagnosis analysis, it was found that *Streptococcus*, *Granulicatella*, and *Leptotrichia* have higher diagnostic value in distinguishing high-risk PN. The combined Area Under Curve (AUC) of three different bacterial genera reached 0.77 (95%CI: 0.62–0.92), suggesting that these three bacterial genera have potential diagnostic significance for distinguishing high-risk PN (Fig. 5).

## Co-occurrence network analysis

The co-occurrence network showed that ASV2, 15, 340, 893, 935, 938, and 3,052 of the LRT microorganisms were common in low-, medium-, and high-risk PN, mainly representing the genera *Veillonella*, *Streptococcus*, *Achromobacter*, *Pseudomonas*, *Pedobacter*, and *Undiabacterium*, while ASV404 (belonging to *Tannerella*) was unique to high-risk PN (Fig. 6A). In the oral microbiota, ASV1, 2, 10, 15, 80, 340, and 899 were common among three groups, mainly representing the genera *Prevotella*, *Veillonella*, *Peptostreptococcus*, *Streptococcus*, *Granulicatella*, and *Rothia* (Fig. 6B), which demonstrated the coexistence of *Veillonella* and *Streptococcus* in the oral and LRT of PN patients, suggesting a potential

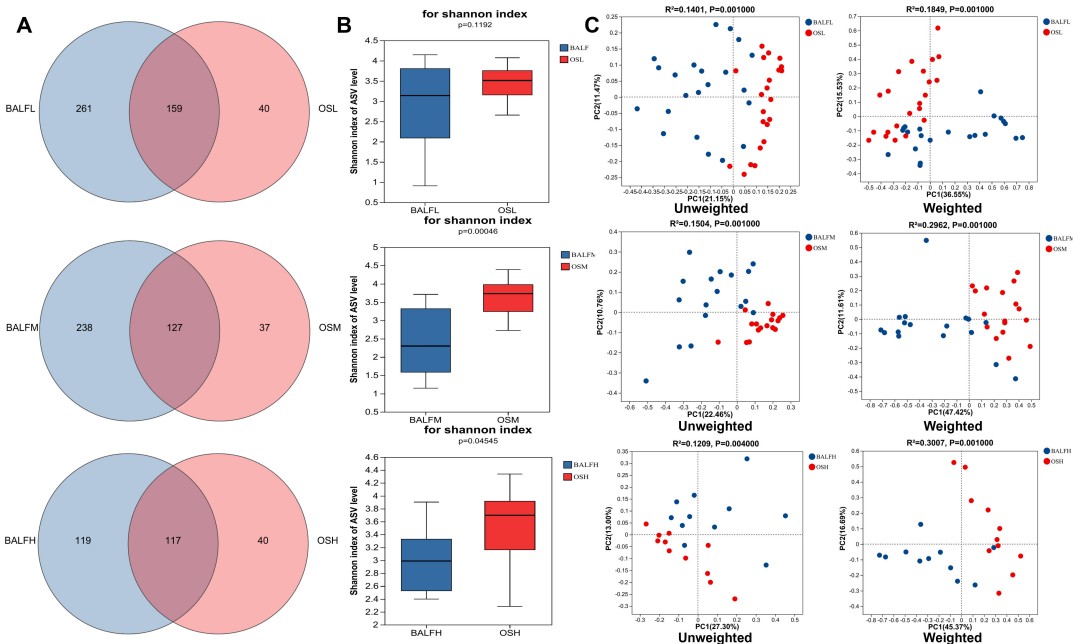

**FIG 4** Comparative analysis of oral microbiota and LRT microbiota. (A) ASV Venn plot. Different colored graphics represent different groups, and the overlapping numbers between different colored graphics represent the number of ASVs shared between two groups. (B) Differences in alpha diversity for Shannon index. Top, low-risk group; middle, medium-risk group; bottom, high-risk group. (C) Beta diversity differences estimated by PCoA. Right, weighted PCoA plots; left, unweighted PCoA plots. BALF group (blue dots); OS group (red dots); top, low-risk group; middle, medium-risk group; bottom, high-risk group; each dot represents a single sample.

association with the occurrence of PN (Fig. 6). In the high-risk group for PN, *Rothia* was significantly enriched in the oral cavity and lower respiratory tract. (Fig. S5C).

Single-factor correlation network analysis was used to forecast the ecological connections among various bacterial communities, focusing on the oral microbiota and LRT microbiota of low-, medium-, and high-risk PN patients (Table S3). A closely linked group of *Veillonella*, including ASV11430, 11432, 10375, 10873, 8400, and 2632, was significantly influential in the bacterial community of the medium- and high-risk samples when compared to the low-risk group. The AUC showed promising discriminatory ability (AUC = 0.72, 95% CI: 0.55–0.89; Fig. S5D). This indicates that *Veillonella* is highly enriched in both the oral cavity and the LRT, which may have a certain correlation with the progression of PN, and further exploration is needed.

A comprehensive analysis was performed to examine the correlation between 13 distinct genera and the 30 most prevalent bacterial genera found in OS and BALF samples across various risk groups for PN (Fig. 7). *Prevotella* and *Veillonella* showed the highest positive correlation ($R^2$ = 0.721, $P$ < 0.001) in the low-risk group, while *Rothia* and *Achromobacter* exhibited the strongest negative correlation ($R^2$ = −0.544, $P$ < 0.001). *Porphyromonas* and *Peptostreptococcus* were the genera with the highest positive correlation in the samples with medium risk, with a correlation coefficient of 0.731 ($P$ < 0.001). *Granulicatella* and *Rothia* were the genera with the highest positive correlation in the samples with high risk, with a correlation coefficient of 0.803 ($P$ < 0.001). In the medium- and high-risk groups, *Prevotella* (R = −0.674, $P$ < 0.001) and *Achromobacter* ($R^2$ = −0.655, $P$ < 0.001) showed the strongest negative correlation (Table S4).

## DISCUSSION

Recent advancements in 16S rRNA sequencing technology have sparked growing curiosity about the potential link between bacteria and various phases of cancer progression, as the LRT microbiota, once thought to be absent, has started to emerge. Notable differences have been observed in the LRT microbiota between LC patients

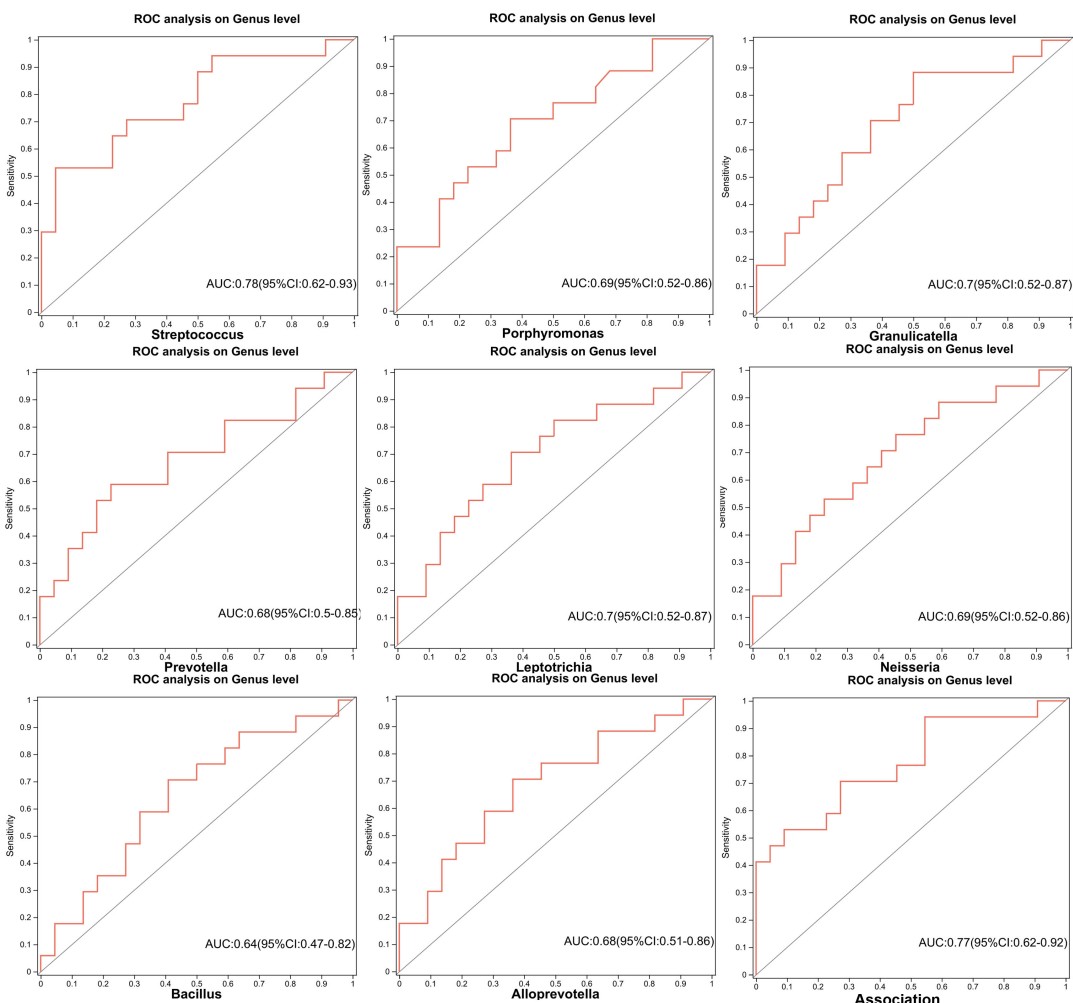

**FIG 5** ROC analysis of eight potential microbiota biomarkers and combinations of the three microorganisms. Association: ROC analysis of *Streptococcus*, *Granulicatella*, and *Leptotrichia*.

and healthy individuals, as well as between LC tissue and normal lung tissue (24–27). Some LRT microbiota can regulate pulmonary inflammatory factors, interfere with cell cycle, upregulate cell proliferation, and promote cancer-related signaling pathways by generating carcinogens, toxins, or interacting with Toll-like receptors on antigen-presenting cells such as monocytes and dendritic cells, thereby promoting the occurrence and development of LC (28). In patients with advanced LC, the LRT microbiota exhibits a composition more akin to that of the oral microbiota than in patients with early-stage LC (9, 21). Similarly, patients with poor survival have a microbiota composition in their LRT samples that is more similar to that of oral samples compared to those with long survival (21). These findings indicate that the enrichment of oral microbiota in the LRT may promote the onset and progression of LC, suggesting that such enriched oral microbiota in the LRT could serve as novel biomarkers for the diagnosis and prognosis of LC.

Consequently, this study specifically examines the oral and lower respiratory microbiota in PN, investigating the relationship between microbial associations in OS and BALF and the malignant transformation of PN. The focus is on identifying biomarkers of oral-lower respiratory symbiotic bacteria that are associated with the risk of malignant PN.

Research has demonstrated that the abundance and diversity of bacterial communities in tumor tissue from LC patients are lower compared to adjacent normal lung tissue (29–31). This observation implies that the unique microenvironment of tumors may

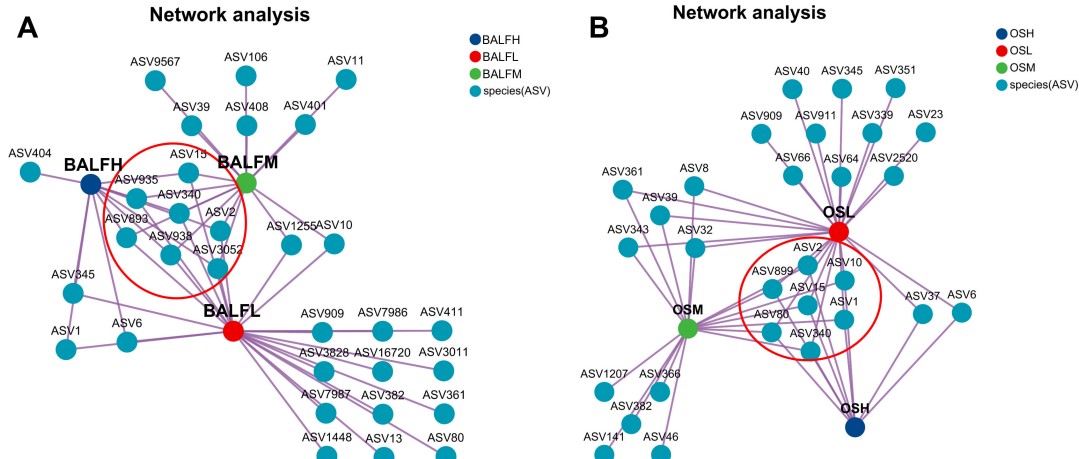

**FIG 6** Co-occurrence network of low-, medium-, and high-risk PN. (A) Co-occurrence network of LRT microbiome. BALFH group (blue dots); BALFL group (red dots); BALFM group (green dots). (B) Co-occurrence network of oral microbiome. OSH group (blue dots); OSL group (red dots); OSM group (green dots). Each node represented an ASV, and the circles indicate the shared dense clusters in both groups.

selectively promote the proliferation of specific bacterial populations (32). Our findings corroborate this, as we observed a reduction in α diversity of the lower airway bacterial microbiome in high-risk PN patients with malignant outcomes. The LEfSe was used to analyze the oral and LRT microbiota of low-, medium-, and high-risk groups, and the results showed that *Synergistes* and *Tannerella* were the most significantly different microbiota. The ROC test was conducted and successfully concluded that both have potential screening significance (Fig. S5A and B). Zhao et al. found that *Synergistes*, as gut microbiota, may be involved in the pathological process of LC (33), possibly through the gut-lung axis by indirectly interacting by regulating immune cells to affect the inflammatory microenvironment of the lung mucosa (34, 35). *Synergistes* and *Tannerella* are closely related to the occurrence of periodontitis (36, 37), which causes severe dysbiosis of the oral and pharyngeal microbiota, increasing the risk of LC by 2.5 times (38). Nevertheless, additional research is required to explore their function in the oral cavity and LRT.

The anatomical configuration of the respiratory tract facilitates the dissemination of oral microbiota to the lower airway (15), thereby contributing to a microbiota

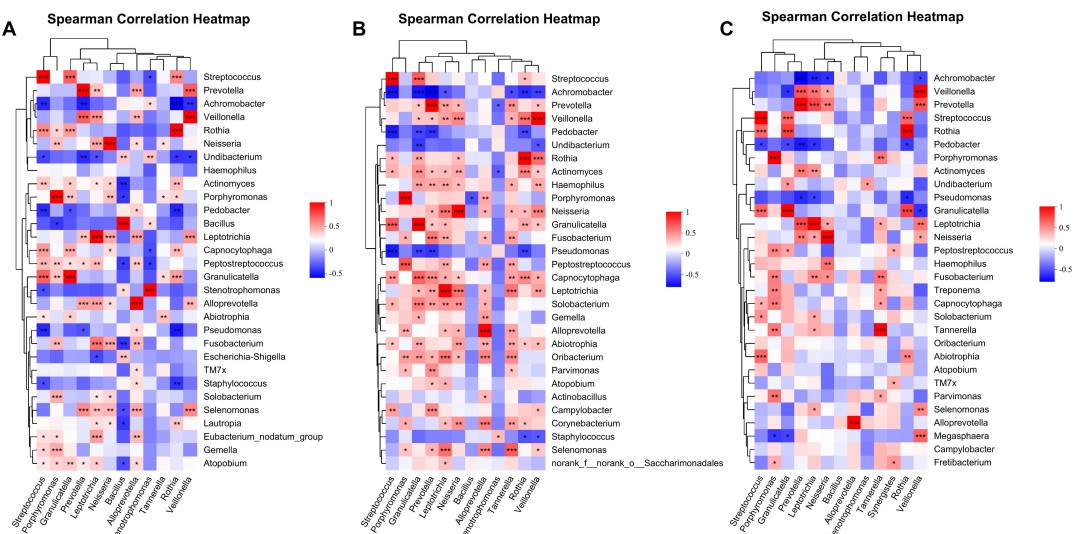

**FIG 7** The analysis of the correlation between 13 differential genera and top 30 most abundant bacterial genera in OS and BALF. (A) Low-risk group. (B) Medium-risk group. (C) High-risk group (*P < 0.05, **P < 0.01, and ***P < 0.001).

composition in the LRT that closely resembles that of the oral cavity (14), particularly in LC patients with poor prognoses (21). Our study revealed that the overlap rate between the LRT microbiota and oral microbiota was 37.9% in low-risk PN patients, 34.8% in medium-risk PN patients, and 49.6% in high-risk PN patients. This result preliminarily indicates that the LRT microbiota of patients with a high risk of PN overlaps with the oral microbiota, which may be related to the enrichment of the oral microbiota in the LRT, which is consistent with previous studies (9, 21). The variation in the abundance of predominant bacterial genera in the oral and LRT flora among low-, medium-, and high-risk groups of pulmonary nodules was examined, identifying eight significant bacterial genera: *Streptococcus*, *Granulicatella*, *Porphyromonas*, *Bacillus*, *Neisseria*, *Alloprevotella*, *Prevotella*, and *Leptotrichia*. Notably, *Streptococcus*, *Granulicatella*, and *Porphyromonas* exhibited no significant differences between the oral cavity and LRT, suggesting that these three genera are enriched in both the oral cavity and LRT.

Simultaneously, *Bacillus*, *Neisseria*, and *Alloprevotella* were identified as significant genera in both the oral cavity and LRT among populations with moderate- and high-risk PN. The aforementioned eight major bacterial genera were validated through ROC analysis, revealing that *Streptococcus*, *Granulicatella*, and *Leptotrichia* may possess potential diagnostic value in distinguishing the malignant risk of PN. Bello et al. found that *Streptococcus* was mainly present in the saliva and bronchi of patients with central LC, which seemed to be the result of oral micro-aspiration leading to the enrichment of the oral microbiome in the bronchi (39). Tsay et al. also found that the LRT of LC patients was enriched with oral microbiota *Streptococcus*, which is related to transcriptome features related to the pathogenesis of LC (21). In addition, Liu et al. (40) found that compared to healthy and adjacent tissues, *Streptococcus* was abundant in LC tumor tissue. *Granulicatella* has previously been identified as a part of the normal respiratory flora (41). Chronic lung inflammation caused by the accumulation of *Granulicatella* may be associated with the etiology of LC (24). Furthermore, tumor staging and metastasis also seem to be associated with the enrichment of the *Granulatella* (42). *Leptotrichia*, a highly prevalent bacteria genus associated with oral cancer (43), has been linked to a reduced risk of LC (44). However, there is insufficient literature to support its enrichment effect in the LRT. Thus, the enrichment of *Streptococcus* and *Granulicatella* in the oral cavity and LRT may be correlated with the malignant transformation of PN. The role of *Leptotrichia* in LC requires further investigation.

Bacteria coexist within complex relational networks, where these interactions significantly influence the species involved, and any disruptions may result in disease. Co-occurrence network analysis revealed an enrichment of *Veillonella* and *Streptococcus* in both the oral cavity and LRT across high-, medium-, and low-risk PN groups. This enrichment suggests a potential association between *Veillonella* and *Streptococcus* and the malignant transformation of PN. Additionally, in high-risk PN cases, *Rothia* was found to be enriched in the oral cavity and LRT. This is consistent with previous studies that found *Rothia* enrichment in the oral and LRT of LC patients (39). In recent years, a number of studies have shown that the abundance of *Veillonella* in various parts of the respiratory tract (such as sputum, OS, and BALF) is highly correlated with LC, and the enrichment of *Veillonella* can promote the occurrence and development of LC (22, 45–47). A recent prospective cohort study provides evidence of a strong association between LRT dysbiological features and LC progression. Subjects with advanced LC were found to have the highest relative abundance of *Veillonella* in the LRT and were associated with poor disease prognosis (22, 45–47). *In vitro* experiments demonstrated that co-culturing *Veillonella* super serum with airway epithelial cells activates the PI3K pathway, upregulates inflammasome-related genes such as IL-17, and promotes the progression of LC, suggesting a potential carcinogenic role for *Veillonella* in LC. Furthermore, animal studies corroborated these findings by showing that the introduction of *Veillonella* into the lungs of lung cancer model mice induces an ecological imbalance in the LRT, triggers severe airway inflammation, increases lung tumor burden, activates the IL-17 inflammatory phenotype and checkpoint inhibitors, and influences the progression

and prognosis of LC (48, 49). Our study identified a network of connections focused on *Veillonella* emerged in the high-risk group of PN, indicating a potential association between *Veillonella* and the development of PN and carcinogenesis. These findings suggest that *Veillonella* should be prioritized in future research to explore its potential role in the malignant transformation of PN.

Several limitations warrant consideration. First, our study only included cases undergoing fiberoptic bronchoscopy examination in the respiratory department of Chengdu Integrated Traditional Chinese and Western Medicine Hospital. Moreover, we acknowledge that we did not conduct the minimum sample size calculation initially. The limited sample size thus affects the reliability and robustness of the research results, potentially leading to bias. Future research should focus on multicenter clinical studies and expand the collection of cases involving high-risk PN patients with definitive pathological diagnoses. Second, we utilized microbiome analysis solely to identify microbial biomarkers related to PN carcinogenesis, yet further validation through metabolomics and genomics data is essential. Finally, underlying lung diseases can impact microbial structures and networks (26), and some collected cases involve patients with chronic obstructive pulmonary disease. In summary, these aspects highlight the need for more comprehensive and rigorous approaches in future investigations to enhance the validity of our findings.

## Conclusion

In summary, we describe for the first time the composition and association of oral and LRT microbiota in patients with PN at different risk of malignancy. Differences in the composition of oral and LRT microbiota were observed among patients with PN exhibiting varying levels of malignant risk. In patients classified within the high-risk group, the oral cavity was predominantly enriched with the bacterial genus *Synergistes*, whereas the LRT showed a significant enrichment of the genus *Tannerella*. Notably, genera such as *Veillonella*, *Streptococcus*, *Granulatella*, *Leptotrichia*, and *Rothia*, which were enriched in both the oral cavity and the LRT, may be correlated with the malignant transformation of PN. This study investigated the correlation between nodule-to-cancer transformation and the enrichment of oral microbiota in the LRT. Additionally, it preliminarily assessed the potential of using oral and LRT microbiota as biomarkers for evaluating the malignancy of PN. The findings offer supportive evidence for the development of microbiota-targeted interventions.

## MATERIALS AND METHODS

### Research methodology

From January 2023 to September 2023, 50 patients with PN who underwent fiberoptic bronchoscopy examination in the Respiratory Department of Chengdu Integrated Traditional Chinese and Western Medicine Hospital were included in this study after preoperative imaging diagnosis. According to the diagnostic guidelines for pulmonary nodules in the Chinese Expert Consensus on Diagnosis and Treatment of Pulmonary Nodules (2018 Edition) (50), PN was defined as round or irregularly shaped spots measuring 3 cm or less in diameter in the lung. These nodules appear as areas of increased density on imaging and may be solitary or multiple, with either distinct or indistinct borders. We used the Mayo model (51), a probability prediction model for PN malignancy, to determine the risk of solid/sub-solid PN malignancy. When the probability of malignancy was ≥0.5, the nodule was at high risk; when the probability of malignancy was <0.5, the nodule was at medium risk; when the probability of malignancy was <0.05, the nodule was at low risk (52). Pure ground-glass nodules and part-solid nodules were determined based on their long diameter (53). The cardiopulmonary function of the subjects was able to withstand bronchoscopy and bronchoalveolar lavage. Subjects had good cardiopulmonary function and were free from active

infections (such as acute pneumonia and pulmonary tuberculosis), oral diseases (such as periodontal inflammation, caries, and oral mucosal diseases), and severe systemic diseases.

Approval for the study was granted by the ethics committee at the Hospital of Chengdu University of Traditional Chinese Medicine and the relevant regulatory agency.

## OS and BALF specimen gathering

All participants agreed to undergo clinical examination and sampling. In-person interviews were carried out with the registered participants to gather fundamental details about weight, height, smoking habits, alcohol intake, past respiratory illnesses, existing health issues, and imaging information (including the size, quantity, density, texture, shape, presence of vascular penetration, calcification, and speculation sign). Non-stimulating OS samples (3–5 mL) were collected with a sterile saliva collector within 1 day prior to bronchoalveolar lavage (54, 55). Subjects were prevented from drinking and eating for at least 90 min before sampling (56). OS samples must be free of impurities like blood, food remnants, and sputum (57, 58) before being preserved in a −80℃ freezer for extended storage. The doctor in the fiberoptic bronchoscopy room followed the principle of "lavage of PN segments" to perform fiberoptic bronchoscopy surgery. We used sterile containers to collect at least 15 mL of each BALF samples, delivered them to the laboratory within 2 hours in a 4℃ transport environment, and frozen and stored them in liquid nitrogen (59).

## DNA extraction

Genomic DNA from the OS and BALF samples was extracted using the E.Z.N.A. kit.soil DNA kit. The genomic DNA was assessed for quality by running it on a 1% agarose gel, and its concentration and purity were measured with NanoDrop2000. The primers that focus on the V3–V4 segment of the 16S ribosomal RNA gene are 338F: 5′-ACTCCT ACGGGAGGCAGCAG′ and 806R: 5′-GGACTACNNGGGTATCTAAT-3′. The PCR amplification protocol began with a pre-denaturation step at 95℃ for 3 minutes, followed by 27 cycles of denaturation at 95℃ for 30 seconds, annealing at 55℃ for 30 seconds, and elongation at 72℃ for 30 seconds. Lastly, there was a final step at 72℃ for 10 minutes. Following the PCR reaction, PCR products from the identical sample were combined and retrieved using a 2% agarose gel, subsequently undergoing identification, purification, and quantification. Ultimately, the library underwent sequencing using the Illumina Miseq PE250 platform. The raw 16S rRNA gene sequences of 50 BALF samples and 50 OS samples were deposited in the NCBI Sequence Read Archive database with accession number PRJNA1116708.

## Sequencing data analysis

The data from each sample were separated from the downstream data using the barcode sequence and PCR extension quotation sequence. FLASH (60) and fastp (61) were then used to merge the reads of each sample. Unqualified sequences were removed after quality filtering (Trimmomatic) to obtain valid data. Effective data quality control measures such as splicing and removal of chimeras were achieved using Uparse algorithm (Uparse v7.0.1001, http://www.drive5.com/uparse/) (62), resulting in 97% consistent ASVs.

By utilizing the SILVA138 bacterium species annotation database from http://www.arb-silva.de/ as the reference sequence and employing the QIIME2 classify sklearn (NaiveBayes) algorithm for ASV annotation, we acquired taxonomic information for ASVs at different levels including kingdom, phylum, class, order, family, genus, and species. The mothur program (https://mothur.org/wiki/mothur_v.1.30.0/) was utilized for the analysis of α diversity, encompassing Chao1, sobs, Shannon, Simpson, and pd index to indicate the diversity and distribution of species in the community. Beta diversity comparison between the three groups were based on the PCoA of weighted UniFrac

and unweighted-uniFrac. The LEfSe analysis was conducted using the LEfSe tool, with a default LDA score threshold of 3.0. Pearson correlation coefficients were calculated to examine the co-occurrence patterns of the top 30 most prevalent taxonomic groups in the samples. Network structures in the bacterial communities of the samples were analyzed using R software (Version 3.3.1) and visualized using Cytoscape (Version 3.9.0). A ROC curve was created in order to identify the specific values of the oral-LRT microbiome that differentiate between individuals with LC and those without. Measurement indicators were compared between two groups using either an independent sample *t*-test for normal distribution or non-parametric test for skewed distribution, while analysis of variance was used for three or more groups with normal distribution or Kruskal-Wallis rank sum test for skewed distribution.

## Statistical analysis

For the statistical analysis of three risk level groups, frequencies at the phylum and genus levels were described. The Kruskal-Wallis H test evaluated differences in alpha diversity indices (Chao1, Sobs, pd, Simpson, and Shannon) (63). If the null hypothesis was rejected, Dunn's test was used for pairwise comparisons (64). We adjusted the *P* value using Bonferroni method to prevent type I error, effectively correcting the alpha value. UniFrac distances were calculated using both weighted (considering sequence abundance) and unweighted (ignoring sequence abundance) methods (65). Beta diversity was evaluated with Adonis analysis (also referred to as Permutational Multivariate Analysis of Variance) (66), and PCoA was used to highlight sample differences (67). LEfSe analysis identified differential species (68), followed by the Kruskal-Wallis H test to assess their abundance levels and Dunn's test for pairwise comparisons. The significance threshold for all tests including post hoc multiple test was $P < 0.05$.

A Venn analysis of ASVs in oral and BALF samples was conducted for patients at low, medium, and high risk for PN. The Wilcoxon rank-sum test assessed alpha diversity differences and 20 main genera between oral and BALF samples across the three risk groups. Beta diversity was compared using the same statistical method as previously described. Abundance variance analysis identified the main bacterial genera in the oral cavity and LRT, while ROC analysis assessed the effectiveness of these bacterial genes in diagnosing high-risk PN.

In the co-occurrence network analysis, single-factor correlation predicted ecological links among bacterial communities, and Spearman correlation assessed relationships between bacterial genera.

## ACKNOWLEDGMENTS

This study was funded by the Young Scientists Fund of the National Natural Science Foundation of China (82305188); Natural Science Foundation of Sichuan Province (23NSFSC1829); China Postdoctoral Science Foundation (2022MD723715); and Foundation of Science and Technology Department of Sichuan Province (2022ZDZX0022).

## AUTHOR AFFILIATIONS

[1]Hospital of Chengdu University of Traditional Chinese Medicine, Jinniu District, Chengdu, Sichuan, China
[2]Cancer Institute, Chengdu University of Traditional Chinese Medicine, Jinniu District, Chengdu, Sichuan, China
[3]Tumor Teaching and Research Office, Chengdu University of Traditional Chinese Medicine, Jinniu District, Chengdu, Sichuan, China

## AUTHOR ORCIDs

Jing Guo http://orcid.org/0000-0001-9861-0250

Yifeng Ren 🔾 http://orcid.org/0000-0003-2976-4250
Xi Fu 🔾 http://orcid.org/0000-0002-0665-6612

## FUNDING

| Funder | Grant(s) | Author(s) |
|---|---|---|
| MOST | National Natural Science Foundation of China (NSFC) | 82305188 | Jing Guo |

## AUTHOR CONTRIBUTIONS

Jing Guo, Conceptualization, Formal analysis, Investigation, Methodology, Validation, Visualization, Writing – original draft | Jierong Han, Formal analysis, Investigation, Methodology, Visualization, Writing – review and editing | Fang Li, Investigation, Methodology, Software, Validation | Qiong Ma, Formal analysis, Investigation, Writing – review and editing | Jiawei He, Investigation, Writing – review and editing | Fengming You, Methodology, Validation | Yifeng Ren, Conceptualization, Methodology, Project administration, Visualization | Xi Fu, Conceptualization, Methodology, Project administration, Supervision, Validation, Visualization

## DATA AVAILABILITY

Sequencing data are available in the NCBI BioProject database (BioProjectID: PRJNA1116708).

## ETHICS APPROVAL

The entire research was authorized by the Ethics Committee of Chengdu University of Traditional Chinese Medicine Affiliated Hospital (approval number: 2022 KL-051). Informed consent was obtained from all participants before their enrollment.

## ADDITIONAL FILES

The following material is available online.

### Supplemental Material

**Figure S1 (Spectrum01284-24-s0001.pdf).** Oral microbiome differences in alpha diversity for chao1 (A), Sobs (B), pd (C), Shannon (D), and Simpson (E) index.
**Figure S2 (Spectrum01284-24-s0002.pdf).** The comparative analysis of differential bacterial genera in oral cavity.
**Figure S3 (Spectrum01284-24-s0003.pdf).** The comparative analysis of differential bacterial genera in LRT.
**Figure S4 (Spectrum01284-24-s0004.pdf).** The association between oral microbiota and LRT microbiota in low-risk PN patients
**Figure S5 (Spectrum01284-24-s0005.pdf).** ROC analysis of (A) oral differential bacteria genus *Synergistes*; (B) LRT differential bacteria genus *Tannerella*; (C) *Rothia</>; (D) Veillonella*.
**Table S1 (Spectrum01284-24-s0006.xlsx).** Basic clinical information of low-, medium-, high-risk pulmonary nodule patients.
**Table S2 (Spectrum01284-24-s0007.xlsx).** Differences in 20 main bacterial genera in oral and lower respiratory tracts.
**Table S3 (Spectrum01284-24-s0008.xlsx).** Single-factor correlation network data for low-, medium-, and high-risk PN group.
**Table S4 (Spectrum01284-24-s0009.xlsx).** The Spearman relationship between 13 biomarkers of microorganisms and top 30 in total abundance.

Open Peer Review

**PEER REVIEW HISTORY (review-history.pdf).** An accounting of the reviewer comments and feedback.

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
