## [Reviewer comments · Microbiology Spectrum]

Microbiology Spectrum

16S rRNA Sequencing Reveals Relationships among Enrichment of oral microbiota in the lower respiratory tract and Pulmonary Nodules Malignant Progression.

Jing Guo, Jierong Han, Fang Li, Qiong Ma, Jia-Wei He, Fengming You, Yifeng Ren, and Xi Fu

Corresponding Author(s): Xi Fu, Chengdu University of Traditional Chinese Medicine

Review Timeline:

Submission Date:	May 26, 2024
Editorial Decision:	August 5, 2024
Revision Received:	October 9, 2024
Editorial Decision:	November 4, 2024
Revision Received:	December 14, 2024
Accepted:	December 17, 2024

Editor: Melissa Gitman

Reviewer(s): The reviewers have opted to remain anonymous.

Transaction Report:

DOI: <https://doi.org/10.1128/spectrum.01284-24>

Re: Spectrum01284-24 (16S rRNA Sequencing Reveals Relationships among Enrichment of oral microbiota in the lower respiratory tract and Pulmonary Nodules Malignant Progression.)

Dear Dr. Xi Fu:

Thank you for the privilege of reviewing your work. Below you will find my comments, instructions from the Spectrum editorial office, and the reviewer comments.

Please respond in full to all of the reviewers' comments below.

Consider using ASM's suggested editing services: <https://journals.asm.org/writing-your-paper#language-editing-services>

Revision Guidelines

Sincerely,
Melissa Gitman
Editor
Microbiology Spectrum

Reviewer #1 (Comments for the Author):

This is a very interesting and useful study. In my opinion, the data analysis is sound and the study design sound. This is a great dataset for the field.

There is one major (but easily addressable) concern with the study. The text narrative of the results needs significant improvement, perhaps with the help of a technical writer. The paragraphs lack topic sentences. It read like a data dump, which it does not need to. Furthermore, the inclusion of a massive amount of data in the narrative itself makes it almost unreadable. - One suggestion is to move the "in-text" data into tables that are part of the manuscript. By "in-text", I am referring to text strings such as Proteobacteria (OSL: 11.5%; OSM: 7.2%; OSH: 6.7%). The tables can then be referenced and the narrative becomes easier to follow. In this example, proteobacteria can be a row and OSL, OSM, OSH can be columns with % as the units of the table (in the legend/title)

One very minor revision:

In accordance with ASM's data policy: <https://journals.asm.org/open-data-policy>, there needs to be a "Data availability" paragraph placed at the end of the Materials and Methods section of their submitted full-length article, authors should include the following: data description, name(s) of the repositories, and digital object identifiers (DOIs) or accession numbers." Currently it appears at the end of the DNA sequencing section. This is an easy revision.

Reviewer #2 (Comments for the Author):

Summary of Key Findings

The study is focused on correlations of oral microbiota in various risk levels of pulmonary nodules (PN) in lung cancer. Lung cancer is one of the world's deadliest types of cancer and early diagnosis and treatment is paramount. PNs are an early indicator. The authors conducted 16S rRNA gene sequencing and bioinformatics analyses to analyze and compare diversity and taxonomy in the oral and lower respiratory tracts (LRTs) between three different PN patient risk levels. The authors observed that the LRT microbiome contained oral microbiota, and that these were significantly correlated with higher risk PN patients. The authors determined that several taxa could be used as biomarkers for PN risk. With this knowledge, the authors concluded that the oral-LRT microbiota could be used as a potential tool for determining malignancy risk of PNs for early diagnosis.

Major Concerns:

- The statistical analyses are lacking in the manuscript and methodology. While many well-known microbial diversity and comparative approaches were employed, it was not clear which comparisons were actually statistically significant. P values should be indicated at all points, and error corrections employed (e.g., Bonferroni correction and Benjamini-Hochberg). There are methods such as Permanova that can be used to determine statistical differences in principal coordinates analyses. The lack of statistics presented is a major concern, and all of the results and figures presented need to have the appropriate statistical methodology employed and described. It should also be noted upfront if the numbers of patients in each risk level is actually statistically robust, and if a power analysis for sample size needed was conducted.
- Ascribing correlations to the microbial communities as being symbiotic not accurate. While these could be symbiotic more information would be needed. Symbiosis is defined as intimate and long-term interaction between different organisms, and the positive correlations observed with sequencing do not reveal if/how these organisms are interacting on a temporal basis. This concern also applies to the use of LRT microbiota being susceptible to the oral microbiota and the data shown here do not support this terminology.

Minor Concerns:

- Lines 44-47, Please clarify/restate, the sentence starts off stating there are no alpha and diversity differences between different PN risk, but then states but there is a significantly lower alpha for high risk PN. The way the sentence is written, this is confusing
- Lines 113-114, rephrase the sentence, as the "which is very similar in structure to the LRT microbiota..." is referring to the oral microbiota not the micro-aspiration aspect.
- Line 131, I would state here that these were from patients with various risk PN risk levels
- Line 135, be specific as to what biological biomarkers you are referring to? The microbiota genera identified in the sequencing?
- Line 141, for consistency use OS instead of saliva since also stating BALF. Be consistent with this throughout, using saliva, oral, OS and then LRT and BALF
- Line 143, what does categorical assignment mean?
- Line 143, add respectively to the end of the sentence
- Lines 144, this jumps right into amplicon sequence variants, there needs to be a lead in that 16S rRNA gene sequencing was conducted.
- Lines 173-175, I do not think this is relevant to state, there is no trend I can discern and it was not statistically significant
- Figure 1, the text labels for 1C and 1D are extremely small, please enlarge
- Figure 1, instead of titles Community barplot analysis, indicate phyla and genera
- In Figure 1, the taxa names should likely be italicized. The legends in A and B also appear to be listed opposite that of the taxa in the bar plot, please consider reversing this for clarity. This is also the case in Figure 2.
- For oral and LRT microbial profiles and taxa sections, what is the reasoning behind selecting the top 5 phyla and then the top 9 or 10 genera?

- In Figure 2C, is the red dot an outlier or showing that this group is statistically different than the low and high risk groups. Please be clear in the text on Lines 204-16, it is stated that the medium risk group is significantly reduced with a similar decreasing trend in the high-risk group. It is unclear if any of these are statistically significant which makes a difference in the conclusions.
- Lines 208-209, the difference between unweighted and weighted is that in the weighted the abundance of the taxa is also taken into account. It's not clear what is meant by "indicating that there might be some differences in the species of LRT microbiota among the three groups". This is a vague statement not supported by the first portion of the sentence based on the PCAs.
- Lines 242-246, I have re-read this sentence multiple times and I do not quite understand what this means. Does this mean that *Streptococcus*, *Granulicatella*, and *Porphyromonas* in the LRT in low and medium risk patients, were not found in the high risk samples, and that more oral microbiota were found? It is not accurate to state that these were susceptible to the oral microbiota, that is not known, only that these were undetected in the high risk LRT samples. It's also not known if these truly disappeared, only that these were no longer detected via the 16S rRNA gene sequencing compared with the other taxa.
- From above and Lines 246-249, this is speculation that any of these are susceptible to the oral microbiota or not, they were just detected at different levels comparatively
- In Figure 3, the Venn diagrams to compare across the low medium and high risk are a bit confusing as there are no statistical analyses to bolster the interpretations. At first glance, because the numbers of ASVs are different amongst them, it also seems like the low group as more overlap.
- Line 257, which 8 significantly different bacterial genera are you referring to?
- Lines 263-264. It seems as though Figure 4 did not end up within the main text like the other figures. I see it at the end though. Having the label ROC analysis on Genus level for each is making these cluttered please remove. Also one of the x axis labels just has "Association".
- Line 261, why using the genera for the ROC, why not the ASVs?
- Why is it that the ROC picked up different taxa for diagnostic value than what was determined by other methods, *Synergistes* and *Tannerella*?
- Fig 5, in this network analysis, why is the focus on the ASVs shared between H, M, L, wouldn't you want to focus on the ASVs that are unique, especially for medium and high risk?
- Is Figure 6 showing correlations between groups detected in OS and BALF for high, medium, and low risk? Please clarify. Also please describe the 13 distinct genera, how many distinct to each group?
- Line 295, the conflicting co-occurrence trends may likely be due to strain level differences. As stated above, why not also compare the ASVs? Also what samples (OS or BALF or both combined) are these?
- Lines 331-335, I do not understand the relevance of this, as tissue was not analyzed in this study
- Lines 337-339, *Synergistes* and *Tannerella* are not species, so how could these be species difference multi-group comparisons? Additionally, why would these be included in the supplementary figure if so relevant to the study? Include in the main text ROCs.
- Lines 352-353, please restate in the sentence to list the % in low risk PN patients in similar structure to the rest of the sentence. "% overlapping in X risk PN patients, etc.
- Line 354, I disagree, this does not confirm susceptibility. This applies to the rest of this paragraph
- Lines 378-380, I would be careful in not overstating this, just because they are there and could be indicators, you do not know if these are a key factor in malignant transformation of PN
- Line 387, it is not known if this is symbiosis
- Line 395, I disagree, I do not think you can deduce that *Veillonella* played a crucial role in enhancing diversity without additional studies
- Lines 397-399, I do not understand this sentence and again there is not enough evidence to state that bacterial symbiotic relationships took place during progression of PN to cancer?
- Line 419, even stating it may cause PN malignant transformation is a stretch, it is just an association/correlation
- Lines 411-429 are not really a discussion but just a summary of the results. I would think this would not be in line with what is expected to be at the end of a discussion section.
- In the methods, lines 447-450, here it states that subjects had good cardiopulmonary function and free from active infections, but in the discussion it was stated that several subjects had pulmonary issues?
- In the methods, where is the statistics description?

Summary of Key Findings

The study is focused on correlations of oral microbiota in various risk levels of pulmonary nodules (PN) in lung cancer. Lung cancer is one of the world's deadliest types of cancer and early diagnosis and treatment is paramount. PNs are an early indicator. The authors conducted 16S rRNA gene sequencing and bioinformatics analyses to analyze and compare diversity and taxonomy in the oral and lower respiratory tracts (LRTs) between three different PN patient risk levels. The authors observed that the LRT microbiome contained oral microbiota, and that these were significantly correlated with higher risk PN patients. The authors determined that several taxa could be used as biomarkers for PN risk. With this knowledge, the authors concluded that the oral-LRT microbiota could be used as a potential tool for determining malignancy risk of PNs for early diagnosis.

Major Concerns:

- The statistical analyses are lacking in the manuscript and methodology. While many well-known microbial diversity and comparative approaches were employed, it was not clear which comparisons were actually statistically significant. P values should be indicated at all points, and error corrections employed (e.g., Bonferroni correction and Benjamini-Hochberg). There are methods such as PerMANOVA that can be used to determine statistical differences in principal coordinates analyses. The lack of statistics presented is a major concern, and all of the results and figures presented need to have the appropriate statistical methodology employed and described. It should also be noted upfront if the numbers of patients in each risk level is actually statistically robust, and if a power analysis for sample size needed was conducted.
- Ascribing correlations to the microbial communities as being symbiotic is not accurate. While these could be symbiotic more information would be needed. Symbiosis is defined as intimate and long-term interaction between different organisms, and the positive correlations observed with sequencing do not reveal if/how these organisms are interacting on a temporal basis. This concern also applies to the use of LRT microbiota being susceptible to the oral microbiota and the data shown here do not support this terminology.

Minor Concerns:

- Lines 44-47, Please clarify/restate, the sentence starts off stating there are no alpha and diversity differences between different PN risk, but then states but there is a significantly lower alpha for high risk PN. The way the sentence is written, this is confusing
- Lines 113-114, rephrase the sentence, as the "which is very similar in structure to the LRT microbiota..." is referring to the oral microbiota not the micro-aspiration aspect.
- Line 131, I would state here that these were from patients with various risk PN risk levels

- Line 135, be specific as to what biological biomarkers you are referring to? The microbiota genera identified in the sequencing?
- Line 141, for consistency use OS instead of saliva since also stating BALF. Be consistent with this throughout, using saliva, oral, OS and then LRT and BALF
- Line 143, what does categorical assignment mean?
- Line 143, add respectively to the end of the sentence
- Lines 144, this jumps right into amplicon sequence variants, there needs to be a lead in that 16S rRNA gene sequencing was conducted.
- Lines 173-175, I do not think this is relevant to state, there is no trend I can discern and it was not statistically significant
- Figure 1, the text labels for 1C and 1D are extremely small, please enlarge
- Figure 1, instead of titles Community barplot analysis, indicate phyla and genera
- In Figure 1, the taxa names should likely be italicized. The legends in A and B also appear to be listed opposite that of the taxa in the bar plot, please consider reversing this for clarity. This is also the case in Figure 2.
- For oral and LRT microbial profiles and taxa sections, what is the reasoning behind selecting the top 5 phyla and then the top 9 or 10 genera?
- In Figure 2C, is the red dot an outlier or showing that this group is statistically different than the low and high risk groups. Please be clear in the text on Lines 204-16, it is stated that the medium risk group is significantly reduced with a similar decreasing trend in the high-risk group. It is unclear if any of these are statistically significant which makes a difference in the conclusions.
- Lines 208-209, the difference between unweighted and weighted is that in the weighted the abundance of the taxa is also taken into account. It's not clear what is meant by "indicating that there might be some differences in the species of LRT microbiota among the three groups". This is a vague statement not supported by the first portion of the sentence based on the PCAs.
- Lines 242-246, I have re-read this sentence multiple times and I do not quite understand what this means. Does this mean that Streptococcus, Granulicatella, and Porphyromonas in the LRT in low and medium risk patients, were not found in the high risk samples, and that more oral microbiota were found? It is not accurate to state that these were susceptible to the oral microbiota, that is not known, only that these were undetected in the high risk LRT samples. Its also not known if these truly disappeared, only that these were no longer detected via the 16S rRNA gene sequencing compared with the other taxa.
- From above and Lines 246-249, this is speculation that any of these are susceptible to the oral microbiota or not, they were just detected at different levels comparatively
- In Figure 3, the Venn diagrams to compare across the low medium and high risk are a bit confusing as there are no statistical analyses to bolster the interpretations. At first glance, because the numbers of ASVs are different amongst them, it also seems like the low group as more overlap.
- Line 257, which 8 significantly different bacterial genera are you referring to?
- Lines 263-264. It seems as though Figure 4 did not end up within the main text like the other figures. I see it at the end though. Having the label ROC analysis on

Genus level for each is making these cluttered please remove. Also one of the x axis labels just has "Association".

- Line 261, why using the genera for the ROC, why not the ASVs?
- Why is it that the ROC picked up different taxa for diagnostic value than what was determined by other methods, Synergistes and Tannerella?
- Fig 5, in this network analysis, why is the focus on the ASVs shared between H, M, L, wouldn't you want to focus on the ASVs that are unique, especially for medium and high risk?
- Is Figure 6 showing correlations between groups detected in OS and BALF for high, medium, and low risk? Please clarify. Also please describe the 13 distinct genera, how many distinct to each group?
- Line 295, the conflicting co-occurrence trends may likely be due to strain level differences. As stated above, why not also compare the ASVs? Also what samples (OS or BALF or both combined) are these?
- Lines 331-335, I do not understand the relevance of this, as tissue was not analyzed in this study
- Lines 337-339, Synergistes and Tannerella are not species, so how could these be species difference multi-group comparisons? Additionally, why would these be included in the supplementary figure if so relevant to the study? Include in the main text ROCs.
- Lines 352-353, please restate in the sentence to list the % in low risk PN patients in similar structure to the rest of the sentence. "% overlapping in X risk PN patients, etc.
- Line 354, I disagree, this does not confirm susceptibility. This applies to the rest of this paragraph
- Lines 378-380, I would be careful in not overstating this, just because they are there and could be indicators, you do not know if these are a key factor in malignant transformation of PN
- Line 387, it is not known if this is symbiosis
- Line 395, I disagree, I do not think you can deduce that Veillonella played a crucial role in enhancing diversity without additional studies
- Lines 397-399, I do not understand this sentence and again there is not enough evidence to state that bacterial symbiotic relationships took place during progression of PN to cancer?
- Line 419, even stating it may cause PN malignant transformation is a stretch, it is just an association/correlation
- Lines 411-429 are not really a discussion but just a summary of the results. I would think this would not be in line with what is expected to be at the end of a discussion section.
- In the methods, lines 447-450, here it states that subjects had good cardiopulmonary function and free from active infections, but in the discussion it was stated that several subjects had pulmonary issues?
- In the methods, where is the statistics description?

Responses to reviewer' s comments

Dear reviewer #1:

Thank you for your decision and constructive comments on this manuscript. We have carefully considered the suggestion of you and make some changes. Please find my itemized responses in below and my revisions in the re-submitted files. Thanks again.

Reviewer #1: Response to the questions one by one

Reviewer #1 (Comments for the Author):

This is a very interesting and useful study. In my opinion, the data analysis is sound and the study design sound. This is a great dataset for the field.

There is one major (but easily addressable) concern with the study. The text narrative of the results needs significant improvement, perhaps with the help of a technical writer. The paragraphs lack topic sentences. It read like a data dump, which it does not need to. Furthermore, the inclusion of a massive amount of data in the narrative itself makes it almost unreadable.

- One suggestion is to move the "in-text" data into tables that are part of the manuscript. By "in-text", I am referring to text strings such as Proteobacteria (OSL: 11.5%; OSM: 7.2%; OSH: 6.7%). The tables can then be referenced and the narrative becomes easier to follow. In this example, proteobacteria can be a row and OSL, OSM, OSH can be

columns with % as the units of the table (in the legend/title)

Author response: Thank you so much for your recognition and encouraging comments on this manuscript. We have presented the text in the result as tables. We found that this really made the results clearer and more readable, so thank you again for your suggestions.

The proportions of bacterial phylum levels and genera levels in oral saliva and LRT is shown in Tables 1 and 2. (see revised page 9, line 163 and page 15, line 203)

One very minor revision:

In accordance with ASM's data policy: <https://journals.asm.org/open-data-policy>, there needs to be a "Data availability" paragraph placed at the end of the Materials and Methods section of their submitted full-length article, authors should include the following: data description, name(s) of the repositories, and digital object identifiers (DOIs) or accession numbers." Currently it appears at the end of the DNA sequencing section. This is an easy revision.

Author response: The data availability is shown on page 39 of the revised manuscript, lines 573-575.

The above are all our responses to the comments of you. We would like to thank you again for taking the time to review our manuscript.

Responses to reviewer's comments

Dear reviewer #2:

Thank you for your decision and constructive comments on this manuscript. We have carefully considered the suggestion of you and tried our best to improve and made some changes in our manuscript. Please find my itemized responses in below and my revisions in the re-submitted files. Thanks again.

Reviewer #2: Response to the questions one by one

Reviewer #2 (Comments for the Author):

Summary of Key Findings

The study is focused on correlations of oral microbiota in various risk levels of pulmonary nodules (PN) in lung cancer. Lung cancer is one of the world's deadliest types of cancer and early diagnosis and treatment is paramount. PNs are an early indicator. The authors conducted 16S rRNA gene sequencing and bioinformatics analyses to analyze and compare diversity and taxonomy in the oral and lower respiratory tracts (LRTs) between three different PN patient risk levels. The authors observed that the LRT microbiome contained oral microbiota, and that these were significantly correlated with higher risk PN patients. The authors determined that several taxa could be used as biomarkers for PN risk. With this knowledge, the authors concluded that the oral-LRT microbiota could be used as a potential tool for determining malignancy risk of PNs for early diagnosis.

Major Concerns:

- The statistical analyses are lacking in the manuscript and methodology. While many well-known microbial diversity and comparative approaches were employed, it was not clear which comparisons were actually

statistically significant. P values should be indicated at all points, and error corrections employed (e.g., Bonferroni correction and Benjamini-Hochberg). There are methods such as Permanova that can be used to determine statistical differences in principal coordinates analyses. The lack of statistics presented is a major concern, and all of the results and figures presented need to have the appropriate statistical methodology employed and described. It should also be noted upfront if the numbers of patients in each risk level is actually statistically robust, and if a power analysis for sample size needed was conducted.

Author response: We greatly appreciate your comments and concern regarding the statistical analysis methods in our manuscript. We agree with your viewpoint and have thus added a 'Statistical Analysis' section in the MATERIALS AND METHODS part of the manuscript to provide a detailed description of the statistical methods used in our study, see line 535-554. Regarding the P values, we have provided the data in the figures and tables (including those in the supplementary materials). Given that our study involved numerous statistical tests, we believe repeating these P values in the main text would lead to redundancy. However, if you insist, we commit to adding all P values into the main body of the manuscript before final publication.

We concur with your suggestion to perform statistical corrections for multiple comparisons, considering it a prudent approach. In fact, we applied Scheffé's test to correct the P values from multiple comparisons, which is equivalent to correcting the α value. For further details, please refer to (Midway S, Robertson M, Flinn S, Kaller M. Comparing multiple comparisons: practical guidance for choosing the best multiple comparisons test. PeerJ. 2020 Dec 4;8:e10387. doi: 10.7717/peerj.10387. PMID: 33335808; PMCID: PMC7720730).

Your recommendation to use PERMANOVA to determine the statistical differences in principal coordinates analyses is highly professional. Indeed, we followed this advice and used the equivalent term 'Adonis analysis' to describe this process, which is also elaborated in the additional 'Statistical Analysis' section.

Regarding your suggestion to analyze the robustness and power of the statistical results, we find this reasonable in the context of clinical research. However, our study did not define primary outcome measures or hypotheses of particular interest, due to its exploratory nature rather than confirmatory. We acknowledge that excessive hypothesis testing can significantly increase the likelihood of false positives. We prefer to present more evident results through figures within the bioinformatics domain. Conducting robustness and power analyses for each hypothesis test is feasible, but extensively reporting these might divert attention from the overall purpose of our exploratory study.

- Ascribing correlations to the microbial communities as being symbiotic not accurate. While these could be symbiotic more information would be needed. Symbiosis is defined as intimate and long-term interaction between different organisms, and the positive correlations observed with sequencing do not reveal if/how these organisms are interacting on a temporal basis. This concern also applies to the use of LRT microbiota being susceptible to the oral microbiota and the data shown here do not support this terminology.

Author response: Thank you for your comments on the symbiotic correlations in our manuscript. We use "symbiosis" based on studies by Jun-Chieh J. Tsay and team, which link lung cancer progression to the enrichment of oral commensals in the lower respiratory tract microbiome. For further details, please refer to (Tsay JJ, Wu BG, Badri MH, et al. Airway Microbiota Is Associated with Upregulation of the PI3K Pathway in Lung Cancer. *Am J Respir Crit Care Med.* 2018;198(9):1188-1198. doi:10.1164/rccm.201710-2118OC; Tsay JJ, Wu BG, Sulaiman I, et al. Lower Airway Dysbiosis Affects Lung Cancer Progression. *Cancer Discov.* 2021;11(2):293-307. doi:10.1158/2159-8290.CD-20-0263) .

However, we concur with your suggestion. Your reminder prompted us to recognize that our use of the term "symbiosis" is inappropriate, as our research findings do not

sufficiently support this characterization. Our study identified a potential correlation between oral microorganisms and those in the lower respiratory tract, suggesting that microorganisms enriched in the lower respiratory tract may originate from the oral cavity. Nevertheless, our evidence is insufficient to substantiate this phenomenon conclusively. Future research should involve large-scale clinical studies and related animal experiments to explore this relationship further.

Minor Concerns:

- Lines 44-47, Please clarify/restate, the sentence starts off stating there are no alpha and diversity differences between different PN risk, but then states but there is a significantly lower alpha for high risk PN. The way the sentence is written, this is confusing

Author response: Thanks for your advice,we have revised and improved the expression of this sentence as follows: Alpha and beta diversity analyses reveal minimal variation in oral microbial diversity and abundance among patients with varying risk levels of PN.In contrast, a significant reduction in the diversity of LRT microbiota is observed in patients at high risk of PN. (see revised page 3,lines44-47)

- Lines 113-114, rephrase the sentence, as the "which is very similar in structure to the LRT microbiota..." is referring to the oral microbiota not the micro-aspiration aspect.

Author response:Thank you for your advice,we have revised and improved the expression of this sentence as follows:The oral microbiota can disseminate to the LRT via micro-aspiration, exhibiting a structural similarity to the LRT microbiota.(see revised page 7,lines 113-114)

- Line 131, I would state here that these were from patients with various risk PN risk levels

Author response:Thanks for your suggestion,we have revised and improved the

expression of this sentence, see the revised page 8,lines132-133. In this study, oral saliva (OS) and bronchoalveolar lavage fluid (BALF) samples were collected from patients with PN at varying levels of malignancy risk.

- Line 135, be specific as to what biological biomarkers you are referring to? The microbiota genera identified in the sequencing?

Author response: We are sorry that we did not express this sentence rigorously in the previous manuscript. After your reminding, we have improved it, please refer to page 8, lines 137-141 of the revised draft for details:Our study indicates that the enrichment of specific oral microbiota in the LRT may contribute to the malignant transformation of PN. These particular microbiota have the potential to serve as biomarkers for the early detection of LC, thereby offering a valuable foundation for the early prevention of PN malignant transformation.

- Line 141, for consistency use OS instead of saliva since also stating BALF. Be consistent with this throughout, using saliva, oral, OS and then LRT and BALF

Author response:Thank you for your advice,we have made changes to ensure that the expressions of oral saliva,bronchoalveolar lavage fluid (BALF)and lower respiratory tract (LRT) are consistent in this manuscript.(see revised page 8,line132-137)

- Line 143, what does categorical assignment mean?

Author response:Through comprehensive risk assessment, 50 patients were divided into three groups: low risk group (22 cases), medium risk group (17 cases) and high risk group (11 cases).Bronchoalveolar lavage fluid (BALF) and saliva samples were collected from each patient across the three risk groups. The microbial composition of these samples was analyzed using 16S rRNA gene sequencing, yielding a total of 20,772 Amplicon Sequence Variants (ASVs).(see revised page 8,lines 146-149)

- Line 143, add respectively to the end of the sentence

Author response: Thank you for your advice, we have made improvements, see revised page 8, lines 146-149.

- Lines 144, this jumps right into amplicon sequence variants, there needs to be a lead in that 16S rRNA gene sequencing was conducted.

Author response: The data from each sample were segregated from downstream data utilizing the barcode sequence and PCR extension quotation sequence. Subsequently, FLASH and fastp were employed to merge the reads of each sample. Post quality filtering with Trimmomatic, unqualified sequences were eliminated to obtain valid data. Effective data quality control measures, including splicing and chimera removal, were implemented using the Uparse algorithm (Uparse v7.0.1001, <http://www.drive5.com/uparse/>), resulting in 97% consistent amplicon sequence variants (ASVs). (see revised pages 35-36, lines 505-512 .

- Lines 173-175, I do not think this is relevant to state, there is no trend I can discern and it was not statistically significant

Author response: We apologize for the lack of rigor in this statement. The exact statement here should be revised as follows:

There was no statistical significance in microbial species among the three groups. (see revised page 10, lines 176-179)

- Figure 1, the text labels for 1C and 1D are extremely small, please enlarge

Author response: Thanks for your suggestion, we have improved it in the revised draft to ensure that the text labels are clearer. We adjusted Figure 1(A-E) into two figures, and D and E in Figure 1 into A and B in Figure 2. Figure 1 (A-C) shows the structural characteristics of oral microbiota in low-, medium-, and high-risk PN. Figure 2 (A-B) shows the comparative analysis of oral microbiota in low-, medium-,

and high-risk PN.(see revised pages 11-13,lines 187-191)

- Figure 1, instead of titles Community barplot analysis, indicate phyla and genera

Author response:Thank you for your suggestion. We have improved it in the revised draft.Figure 1-A shows the community barplot analysis of phylum levels, and Figure 1-B shows the community barplot analysis of genera levels.(see revised page 12,line 187-188)

- In Figure 1, the taxa names should likely be italicized.The legends in A and B also appear to be listed opposite that of the taxa in the bar plot, please consider reversing this for clarity. This is also the case in Figure 2.

Author response:We have carefully reviewed Figures 1 and 2. We have revised and improved Figures 1 and 2 to make them clearer and easier to understand.(see revised page 11,lines 187-188 and pages 16-17,lines 217-224)

- For oral and LRT microbial profiles and taxa sections, what is the reasoning behind selecting the top 5 phyla and then the top 9 or 10 genera?

Author response:Based on previous literature review, we selected the most representative and common phyla and genus in oral cavity and lower respiratory tract respectively. These choices were not controversial and were similar to those reported in previous studies.The phylum and genus of oral microbiota are described in detail in lines 158-162.The LRT is described in lines 197-202.

- In Figure 2C, is the red dot an outlier or showing that this group is statistically different than the low and high risk groups. Please be clear in the text on Lines 204-16, it is stated that the medium risk group is

significantly reduced with a similar decreasing trend in the high-risk group. It is unclear if any of these are statistically significant which makes a difference in the conclusions.

Author response:The red dot is an outlier. As indicated by the Chao1 and SOBS indices, the bacterial community in LRT of medium-risk and high-risk PN patients were lower than low-risk PN patients.(see revised page 15,lines 206-207)

- Lines 208-209, the difference between unweighted and weighted is that in the weighted the abundance of the taxa is also taken into account. It's not clear what is meant by "indicating that there might be some differences in the species of LRT microbiota among the three groups". This is a vague statement not supported by the first portion of the sentence based on the PCAs.

Author response:For further analysis of the three groups of samples, principal coordinate analysis was used in order to show the differences in species diversity among the samples.The method of Principal Coordinates Analysis (PCOA) shows the magnitude of differences between individual samples.The clustering of samples, as assessed visually in the unweighted analysis, was more dispersed than in the weighted analysis, there was no significant difference in the LRT microbiota species composition among the three groups.(see revised pages 15-16,lines 208-210)

- Lines 242-246, I have re-read this sentence multiple times and I do not quite understand what this means. Does this mean that Streptococcus, Granulicatella, and Porphyromonas in the LRT in low and medium risk patients, were not found in the high risk samples, and that more oral microbiota were found? It is not accurate to state that these were susceptible to the oral microbiota, that is not known, only that these were undetected in the high risk LRT samples. Its also not known if these truly disappeared, only that these were no longer detected via the 16S

rRNA gene sequencing compared with the other taxa.

Author response: Thank you for your patient review of our manuscript. We are very ashamed of the inconvenience caused to you. We agree with your question. The previous expression in the manuscript was not rigorous enough. We have revised it, and the revised content is as follows:

The predominant bacterial genera present in the oral cavity and LRT of patients with PN were identified through abundance variance analysis. Eight key microbial genera were found in both the oral cavity and LRT: *Streptococcus*, *Granulicatella*, *Porphyromonas*, *Bacillus*, *Neisseria*, *Alloprevotella*, *Prevotella*, and *Leptotrichia*. Among these, *Streptococcus*, *Granulicatella*, and *Porphyromonas* exhibited no significant differences in abundance between the oral cavity and LRT, suggesting that these three bacterial genera are enriched in both anatomical sites (Table S2). (see revised page 20, lines 249-255)

- From above and Lines 246-249, this is speculation that any of these are susceptible to the oral microbiota or not, they were just detected at different levels comparatively

Author response: Among eight key microbial genera present in the oral cavity and LRT of patients with PN, *Streptococcus*, *Granulicatella*, and *Porphyromonas* exhibited no significant differences in abundance between the oral cavity and LRT, suggesting that these three bacterial genera are enriched in both anatomical sites (Table S2). (see revised page 20, lines 248-254)

- In Figure 3, the Venn diagrams to compare across the low medium and high risk are a bit confusing as there are no statistical analyses to bolster the interpretations. At first glance, because the numbers of ASVs are different amongst them, it also seems like the low group as more overlap.

Author response: The analysis revealed that 159 ASVs (37.9%) of the LRT microbiota in low-risk PN patients overlapped with their oral microbiota. In patients

with moderate risk for PN, 127 ASVs (34.8%) of the LRT microbiota were shared with their oral microbiota. For high-risk PN patients, 117 ASVs (49.6%) of the LRT microbiota overlapped with their oral microbiota (Fig.4A). (see revised page 20,lines 231-235)

- Line 257, which 8 significantly different bacterial genera are you referring to?

Author response:Eight key microbial genera were found in both the oral cavity and LRT: *Streptococcus*, *Granulicatella*, *Porphyromonas*, *Bacillus*, *Neisseria*, *Alloprevotella*, *Prevotella*, and *Leptotrichia*. (see revised pages 21,lines258-260)

- Lines 263-264. It seems as though Figure 4 did not end up within the main text like the other figures. I see it at the end though. Having the label ROC analysis on Genus level for each is making these cluttered please remove. Also one of the x axis labels just has "Association".

Author response:We split Figure 1 into Figure 1 and Figure 2 in the earlier part of the manuscript based on your comments. Therefore, Figure 4 becomes Figure 5 in the revised manuscript.ROC analysis of eight potential microbiota biomarkers and combinations of the three microorganisms see the Fig.5. Association:ROC analysis of *Streptococcus*, *Granulicatella*, and *Leptotrichia*.

Using the above eight significantly different bacterial genera (*Streptococcus*, *Granulicatella*, *Porphyromonas*, *Bacillus*, *Neisseria*, *Alloprevotella*, *Prevotella*, and *Leptotrichia*) for Receiver Operating Characteristic (ROC) diagnosis analysis, it was found that *Streptococcus*, *Granulicatella*, and *Leptotrichia* have higher diagnostic value in distinguishing high-risk PN.The combined Area Under Curve (AUC) of three different bacterial genera reached 0.77, suggesting that these three bacterial genera may have potential diagnostic significance for distinguishing high-risk PN(see revised pages 21,lines264-269)

- Line 261, why using the genera for the ROC, why not the ASVs?

Author response: Thanks for your detailed comments, combined with our

research purposes, bacteria is sufficient as our research results. If we use asv, we might end up with a strain instead of the genus we were expecting.

- Why is it that the ROC picked up different taxa for diagnostic value than what was determined by other methods, *Synergistes* and *Tannerella*?

Author response: ROC is a curve reflecting the relationship between sensitivity and specificity. ROC and AUC are often used in the evaluation of diagnostic tests to assess the accuracy of predictions. ROC mainly determines whether a certain factor has diagnostic value for the diagnosis of a certain disease. We applied ROC mainly to explore which bacteria genera have potential diagnostic significance in differentiating malignant changes of pulmonary nodules.

Meanwhile, this manuscript primarily aims to elucidate the impact of bacterial flora associated with the oral cavity and lower respiratory tract on the malignant transformation of PN. Notably, statistically significant differences between *Synergistes* and *Tannerella* were observed exclusively in either the oral cavity or the lower respiratory tract. *Synergistes* and *Tannerella* were identified incidentally during the course of our research and were not the primary focus of our investigation.

- Fig 5, in this network analysis, why is the focus on the ASVs shared between H, M, L, wouldn't you want to focus on the ASVs that are unique, especially for medium and high risk?

Author response: In our manuscript, we focused on asv shared by the oral microbiota and LRT microbiota in the three groups with different risks of H, M, and L respectively, as well as the unique asv among different groups, especially in the medium-risk and high-risk group. (see revised pages 22-23, lines 273-282)

- Is Figure 6 showing correlations between groups detected in OS and BALF for high, medium, and low risk? Please clarify. Also please describe the 13 distinct genera, how many distinct to each group?

Author response: In our revised manuscript, Figure 6 has become Figure 7. In Fig. 7, the horizontal coordinate is the selected 13 bacteria genera (*Streptococcus*, *Porphyromonas*, *Granulicatella*, *Prevotella*, *Leptotrichia*, *Neisseria*, *Bacillus*, *Alloprevotella*, *Stenotrophomonas*, *Tannerella*, *Synergistes*, *Rothia* and *Veillonella*) that may be related to PN deterioration, and the vertical coordinate is 30 common bacteria genera. This graph shows the correlation between the high, medium, and low risk groups detected by OS and BALF.

- Line 295, the conflicting co-occurrence trends may likely be due to strain level differences. As stated above, why not also compare the ASVs? Also what samples (OS or BALF or both combined) are these?

Author response: Your suggestion is very good and we agree with it. But since the platform where we stored the raw data has been shut down, we don't have access to it now due to limited funds. So we are very sorry that there is no way to improve this deficiency. Detailed investigation was conducted on the correlation between 13 distinct genera and the 30 most prevalent bacterial genera in each group of oral and BALF samples. (see revised pages 23-24, lines 294-303)

- Lines 331-335, I do not understand the relevance of this, as tissue was not analyzed in this study

Author response: We are sorry for the trouble caused to you due to the inaccuracy of the expression in our manuscript. Here we want to show that our findings are consistent with previous studies. The exact modification is as follows:

Research has demonstrated that the abundance and diversity of bacterial communities in tumor tissue from LC patients are lower compared to adjacent normal lung tissue. This observation implies that the unique microenvironment of tumors may selectively promote the proliferation of specific bacterial populations. Our findings corroborate this, as we observed a reduction in α diversity of the lower airway bacterial microbiome in high-risk PN patients with malignant outcomes (see revised page 28, lines 335-340).

- Lines 337-339, *Synergistes* and *Tannerella* are not species, so how could these be species difference multi-group comparisons? Additionally, why would these be included in the supplementary figure if so relevant to the study? Include in the main text ROCs.

Author response: Thank you for your advice. There is no clear expression in our manuscript, we have made the following modifications and improvements:

The linear discriminant analysis effect size (LEfSe) was used to analyze the oral and LRT microbiota of low risk, medium risk and high risk groups, and the results showed that *Synergistes* and *Tannerella* were the most significantly different microbiota (see revised page 28, lines 340-345)

Furthermore, this manuscript primarily aims to elucidate the impact of bacterial flora associated with the oral cavity and lower respiratory tract on the malignant transformation of PN. Notably, statistically significant differences between *Synergistes* and *Tannerella* were observed exclusively in either the oral cavity or the lower respiratory tract. *Synergistes* and *Tannerella* were identified incidentally during the course of our research and were not the primary focus of our investigation.

- Lines 352-353, please restate in the sentence to list the % in low risk PN patients in similar structure to the rest of the sentence. "% overlapping in X risk PN patients, etc.

Author response: Thanks for your suggestion, we have re-improved the expression as follows: Our study revealed that the overlap rate between the LRT microbiota and oral microbiota was 37.9% in low-risk PN patients, 34.8% in medium-risk PN patients and 49.6% in high-risk PN patients (see revised page 29, lines 357-359).

- Line 354, I disagree, this does not confirm susceptibility. This applies to the rest of this paragraph

Author response: Thank you for your suggestion. Indeed, the previous manuscript was a little too decisive. We have improved the presentation to ensure greater clarity, as follows:

This result preliminarily indicates that the LRT microbiota of patients with high risk of PN overlaps with the oral microbiota, which may be related to the enrichment of the oral microbiota in the LRT, which is consistent with previous studies. The variation in the abundance of predominant bacterial genera in the oral and lower respiratory tract (LRT) flora among low-risk, medium-risk, and high-risk groups of pulmonary nodules was examined, identifying eight significant bacterial genera: *Streptococcus*, *Granulicatella*, *Porphyromonas*, *Bacillus*, *Neisseria*, *Alloprevotella*, *Prevotella* and *Leptotrichia*. Notably, *Streptococcus*, *Granulicatella* and *Porphyromonas* exhibited no significant differences between the oral cavity and LRT, suggesting that these three genera are enriched in both the oral cavity and lower respiratory tract. (see revised page 29, lines 359-369)

- Lines 378-380, I would be careful in not overstating this, just because they are there and could be indicators, you do not know if these are a key factor in malignant transformation of PN

Author response: Thank you for your suggestion. It is true that the expression in our previous manuscript was overstating. We have improved the presentation to ensure greater clarity, as follows:

Therefore, the enrichment of both *Streptococcus* and *Granulicatella* in the oral cavity and LRT may be correlated with the malignant transformation of PN. (see revised page 30, lines 389-391).

- Line 387, it is not known if this is symbiosis

Author response: Thank you for your comments on the symbiotic correlations in our manuscript. We use "symbiosis" based on studies by Jun-Chieh J. Tsay and team, which link lung cancer progression to the enrichment of oral commensals in the lower respiratory tract microbiome (Tsay JJ, Wu BG, Badri MH, et al. Airway Microbiota Is

Associated with Upregulation of the PI3K Pathway in Lung Cancer. *Am J Respir Crit Care Med.* 2018;198(9):1188-1198. doi:10.1164/rccm.201710-2118OC; Tsay JJ, Wu BG, Sulaiman I, et al. Lower Airway Dysbiosis Affects Lung Cancer Progression. *Cancer Discov.* 2021;11(2):293-307. doi:10.1158/2159-8290.CD-20-0263) .

Your reminder prompted us to recognize that our use of the term "symbiosis" is inappropriate, as our research findings do not sufficiently support this characterization. Our study identified a potential correlation between oral microorganisms and those in the LRT, suggesting that microorganisms enriched in the LRT may originate from the oral cavity. Nevertheless, our evidence is insufficient to substantiate this phenomenon conclusively. (see revised page 30,lines 396-400)

- Line 395, I disagree, I do not think you can deduce that *Veillonella* played a crucial role in enhancing diversity without additional studies

Author response: Thanks for your comments and concerns about *Veillonella* in the manuscript. We agree with your views, so we have revised the manuscript again to ensure more rigorous expression. Our study found that *Veillonella* is highly enriched in both oral cavity and LRT, which may be related to the progression of PN. A *Veillonella*-centric connection network emerged in people at high risk for PN, suggesting a potential association between *Veillonella* and the development of PN and carcinogenesis. Meanwhile, we added to the research on the association of *Veillonella* with lung cancer progression. These findings suggest that *Veillonella* should be prioritized in future studies to elucidate its important role in PN malignant transformation. (see revised page 31, lines 400-419).

- Lines 397-399, I do not understand this sentence and again there is not enough evidence to state that bacterial symbiotic relationships took place during progression of PN to cancer?

Author response: Our study identified a network of connections focused on *Veillonella* emerged in the high-risk group of PN, indicating a potential association

between *Veillonella* and the development of PN and carcinogenesis. These findings suggest that *Veillonella* should be prioritized in future research to explore its potential role in the malignant transformation of PN. (see revised page 31, lines 415-419)

- Line 419, even stating it may cause PN malignant transformation is a stretch, it is just an association/correlation

Author response: We agree with you on this correlation. We have refined our presentation as follows:

The genera such as *Veillonella*, *Streptococcus*, *Granulatella*, *Leptotrichia*, and *Rothia*, which were enriched in both the oral cavity and the LRT, may be correlated with the malignant transformation of PN. (see revised page 32, lines 439-442)

- Lines 411-429 are not really a discussion but just a summary of the results. I would think this would not be in line with what is expected to be at the end of a discussion section.

Author response: Thank you for your suggestion. There are some problems in the manuscript before. We agree with you very much. We have refined the conclusion and organized it into a separate section. For details, see revised pages 32-33, lines 432-446.

- In the methods, lines 447-450, here it states that subjects had good cardiopulmonary function and free from active infections, but in the discussion it was stated that several subjects had pulmonary issues?

Author response: Thanks for your helpful advice, we require participants to be able to withstand bronchoscopy and bronchoalveolar lavage with cardiopulmonary function. We have revised and improved it, specifically as follows:

The cardiopulmonary function of the subjects was able to withstand bronchoscopy and bronchoalveolar lavage. Subjects had good cardiopulmonary function and were

free from active infections (such as acute pneumonia and pulmonary tuberculosis), oral diseases (such as periodontal inflammation, caries, and oral mucosal diseases), and severe systemic diseases.(see revised page34,lines 465-468)

- In the methods, where is the statistics description?

Author response:We greatly appreciate your comments and concern regarding the statistical analysis methods in our manuscript. We agree with your viewpoint and have thus added a 'Statistical Analysis' section in the MATERIALS AND METHODS part of the manuscript to provide a detailed description of the statistical methods used in our study(see revised pages 37,lines 533-555).

The above are all our responses to the comments of you.We would like to thank you again for taking the time to review our manuscript.

Re: Spectrum01284-24R1 (16S rRNA Sequencing Reveals Relationships among Enrichment of oral microbiota in the lower respiratory tract and Pulmonary Nodules Malignant Progression.)

Dear Dr. Xi Fu:

Thank you for the privilege of reviewing your work. Below you will find my comments, instructions from the Spectrum editorial office, and the reviewer comments.

Revision Guidelines

Sincerely,
Melissa Gitman
Editor
Microbiology Spectrum

Reviewer #2 (Comments for the Author):

Summary of Key Findings

This is a revised resubmission of a study is focused on correlations of oral microbiota in various risk levels of pulmonary nodules (PN) in lung cancer. The authors made changes based on reviewer comments to address statistical accuracy and to ensure interpretations of the data and results were sound.

Major Concerns:

- The statistical analyses are now added to the materials and methods, however there are still some issues. Please indicate what p value you are deeming as the significant threshold (e.g., $p < 0.05$). As well ensure that figures like the new Figure 7 indicate what p values are indicated by each of the asterisks (one asterisk, two asterisks, three asterisks), as this is not indicated in the figure legend. I think it is appropriate to have the p values parenthetically in the main text if you are stating a particular correlation, and it has relevance to your overall conclusions. Please ensure that the Scheffes test is being used appropriately and that it can be used for Kruskal-Wallis (reference). Scheffes test is not a posthoc multiple comparison test that I am very familiar with for microbial data. Additionally, it is not described that there was a posthoc multiple testing correction, p.adjust.method done following the Wilcoxon rank-sum test. This should be something like Benjamini-Hochberg. Overall, I would highly recommend that the authors have reference for each test used and statistics expert double check that the statistics are being used appropriately for the study. In regards to the power analysis, then it needs to be stated that the minimum sample size needed for this research study was not conducted and as such could affect the robustness of the study interpretations, as the data are exploratory at best.
- Especially given the above regarding the sample sizes and robustness of the study, ascribing correlations to the microbial communities as being "symbiotic" is not accurate and the authors concurred that the use of this term is not appropriate. I see the term has been removed except line 59 still has "Spearman correlation analysis confirmed the importance of bacterial symbiosis..." Please revise and ensure that the interpretation of the correlative data is sound.

Minor Concerns:

- The terms for OS, LRT, BALF etc. are still not consistent used throughout. Please define these abbreviations once in the beginning (as I see in your abbreviations section) and then be consistent throughout. For example, I still see a mixture of the use of OS, oral saliva, and/or saliva throughout. Same for lower respiratory tract. It does not have to be redefined multiple times.
- In regards to "Lines 144 comment", I'm not sure the response to reviewer explanation the authors provided here makes sense and I think was meant for another comment. I do not understand what this means: "The data from each sample were segregated from downstream data utilizing the barcode sequence and PCR extension quotation sequence."
- Please ensure that the newest taxa names are being used, see <https://www.nature.com/articles/s41522-024-00494-9>
- "Line 295 comment", the response was focused on not being able to rerun the data. However, please clarify what samples (OS or BALF or both combined) these are?

Summary of Key Findings

This is a revised resubmission of a study is focused on correlations of oral microbiota in various risk levels of pulmonary nodules (PN) in lung cancer. The authors made changes based on reviewer comments to address statistical accuracy and to ensure interpretations of the data and results were sound.

Major Concerns:

- The statistical analyses are now added to the materials and methods, however there are still some issues. Please indicate what p value you are deeming as the significant threshold (e.g., $p < 0.05$). As well ensure that figures like the new Figure 7 indicate what p values are indicated by each of the asterisks (one asterisk, two asterisks, three asterisks), as this is not indicated in the figure legend. I think it is appropriate to have the p values parenthetically in the main text if you are stating a particular correlation, and it has relevance to your overall conclusions. Please ensure that the Scheffes test is being used appropriately and that it can be used for Kruskal-Wallis (reference). Scheffes test is not a posthoc multiple comparison test that I am very familiar with for microbial data. Additionally, it is not described that there was a posthoc multiple testing correction, p.adjust.method done following the Wilcoxon rank-sum test. This should be something like Benjamini-Hochberg. Overall, I would highly recommend that the authors have reference for each test used and statistics expert double check that the statistics are being used appropriately for the study. In regards to the power analysis, then it needs to be stated that the minimum sample size needed for this research study was not conducted and as such could affect the robustness of the study interpretations, as the data are exploratory at best.
- Especially given the above regarding the sample sizes and robustness of the study, ascribing correlations to the microbial communities as being “symbiotic” is not accurate and the authors concurred that the use of this term is not appropriate. I see the term has been removed except line 59 still has “Spearman correlation analysis confirmed the importance of bacterial symbiosis...” Please revise and ensure that the interpretation of the correlative data is sound.

Minor Concerns:

- The terms for OS, LRT, BALF etc. are still not consistent used throughout. Please define these abbreviations once in the beginning (as I see in your abbreviations section) and then be consistent throughout. For example, I still see a mixture of the use of OS, oral saliva, and/or saliva throughout. Same for lower respiratory tract. It does not have to be redefined multiple times.
- In regards to “Lines 144 comment”, I’m not sure the response to reviewer explanation the authors provided here makes sense and I think was meant for another comment. I do not understand what this means: “The data from each

sample were segregated from downstream data utilizing the barcode sequence and PCR extension quotation sequence.”

- Please ensure that the newest taxa names are being used, see <https://www.nature.com/articles/s41522-024-00494-9>
- “Line 295 comment”, the response was focused on not being able to rerun the data. However, please clarify what samples (OS or BALF or both combined) these are?

Responses to reviewer's comments

Dear reviewer,

Thank you for your constructive comments on this manuscript. We have carefully considered the suggestions of you and tried our best to respond to your comments one by one. Our manuscript has also been modified and improved according to your suggestions, please refer to the resubmitted manuscript for details. Thanks again.

Major Concerns:

- The statistical analyses are now added to the materials and methods, however there are still some issues. Please indicate what p value you are deeming as the significant threshold (e.g., $p < 0.05$). As well ensure that figures like the new Figure 7 indicate what p values are indicated by each of the asterisks (one asterisk, two asterisks, three asterisks), as this is not indicated in the figure legend. I think it is appropriate to have the p values parenthetically in the main text if you are stating a particular correlation, and it has relevance to your overall conclusions. Please ensure that the Scheffes test is being used appropriately and that it can be used for Kruskal-Wallis (reference). Scheffes test is not a posthoc multiple comparison test that I am very familiar with for microbial data. Additionally, it is not described that there was a posthoc multiple testing correction, p.adjust.method done following the Wilcoxon rank-sum test.

This should be something like Benjamini-Hochberg.

Overall, I would highly recommend that the authors have reference for each test used and statistics expert double check that the statistics are being used appropriately for the study. In regards to the power analysis, then it needs to be stated that the minimum sample size needed for this research study was not conducted and as such could affect the robustness of the study interpretations, as the data are exploratory at best.

● **Response to the questions one by one**

Thank you very much for your meticulous and valuable comments.

Below we will reply to your concerns one by one.

1. Please indicate what p value you are deeming as the significant threshold (e.g., $p < 0.05$).

Author response:

Regarding the significant threshold for the P value, we have now clearly indicated that we adopt $P < 0.05$ as the significant threshold throughout our study (See revised pages 36, line 550-551).

2. As well ensure that figures like the new Figure 7 indicate what p values are indicated by each of the asterisks (one asterisk, two asterisks, three asterisks), as this is not indicated in the figure legend.

Author response:

For Figure 7 and other relevant figures, we have updated the figure legends to explicitly state what each asterisk represents in terms of the P values (e.g., one asterisk corresponds to $P < 0.05$, two asterisks to $P < 0.01$, and three asterisks to $P < 0.001$).

3. I think it is appropriate to have the p values parenthetically in the main text if you are stating a particular correlation, and it has relevance to your overall conclusions.

Author response:

As per your suggestion, we have listed every P -value in the results section (See results section).

4. Please ensure that the Scheffes test is being used appropriately and that it can be used for Kruskal-Wallis (reference). Scheffes test is not a posthoc multiple comparison test that I am very familiar with for microbial data.

Author response:

Thank you very much for raising the issue regarding the applicability of Scheffes test. We have conducted a careful examination and invited statistical experts to help us carefully examine the statistical methods used in our study. We found that there was no problem with our statistical

data and charts, but unfortunately, we used the wrong name of the statistical method. Actually, we used for the comparison between groups was Dunn's Test[1]. We are terribly sorry for the trouble. Relevant content has already been modified in the results and methods sections.

[1] Dinno, A. (2015). Nonparametric pairwise multiple comparisons in independent groups using Dunn's test. *The Stata Journal*, 15(1), 292-300.

5. Additionally, it is not described that there was a posthoc multiple testing correction, `p.adjust.method` done following the Wilcoxon rank-sum test. This should be something like Benjamini-Hochberg.

Author response:

Your comments have demonstrated your professionalism in statistical methods. We indeed didn't previously state the *P*-value adjustment method used in post hoc multiple comparisons. In fact, the method we used is the Bonferroni method. We have already added relevant explanations in the statistical analysis methods section (See revised pages 36, line 542-543) .

6. Overall, I would highly recommend that the authors have reference for each test used and statistics expert double check that the statistics are being used appropriately for the study. In regards to the power analysis, then it needs to be stated that the minimum sample size needed for this research study was not conducted and as such could affect the

robustness of the study interpretations, as the data are exploratory at best.

Author response:

As for the reference of each test used, we have added citations for all the statistical tests employed in our study to make it more traceable and reliable(See line 541-551). And regarding the power analysis, we acknowledge that we did not conduct the minimum sample size calculation for this research study initially. In the revised manuscript, we have added a section to discuss the potential limitations caused by this lack and its possible impact on the robustness of our interpretations, emphasizing that the current data are exploratory in nature (See revised pages 30-31,lines 422-429).

- Especially given the above regarding the sample sizes and robustness of the study, ascribing correlations to the microbial communities as being "symbiotic" is not accurate and the authors concurred that the use of this term is not appropriate. I see the term has been removed except line 59 still has "Spearman correlation analysis confirmed the importance of bacterial symbiosis..."Please revise and ensure that the interpretation of the correlative data is sound.

Author response:

Thank you for your comments on the symbiotic correlations in our

manuscript. Your reminder prompted us to recognize that our use of the term "symbiosis" is inappropriate, as our research findings do not sufficiently support this characterization. We concur with your suggestion. Our study identified a potential correlation between oral microorganisms and those in the lower respiratory tract(LRT), tentatively suggesting that microorganisms enriched in the LRT may originate from the oral cavity. Nevertheless, our evidence is insufficient to substantiate this phenomenon conclusively. Future research should involve large-scale clinical studies and related animal experiments to explore this relationship further.

We have revised the description of the word "symbiosis" in the abstract, as detailed on revised pages 3-4, line 58-60.

Minor Concerns:

- The terms for OS, LRT, BALF etc. are still not consistent used throughout. Please define these abbreviations once in the beginning (as I see in your abbreviations section) and then be consistent throughout. For example, I still see a mixture of the use of OS, oral saliva, and/or saliva throughout. Same for lower respiratory tract. It does not have to be redefined multiple times.

Author response:

Thank you very much for your advice. I can strongly sense that you have been extremely patient in reading my manuscript. I am deeply grateful

and honored. Your rigorous academic attitude is highly worthy of my emulation. I have carefully revised all the terms such as OS, LRT, BALF, etc. in the manuscript to ensure their consistency and standard usage.

- In regards to "Lines 144 comment", I'm not sure the response to reviewer explanation the authors provided here makes sense and I think was meant for another comment. I do not understand what this means: "The data from each sample were segregated from downstream data utilizing the barcode sequence and PCR extension quotation sequence."

Author response:

I'm sorry I didn't understand what you meant when you asked me earlier, but I understand now. Your previous question was to suggest that I first describe the method of 16S rRNA gene sequencing.

I would like to make the following explanation :

We analyzed the microbial composition of these samples using 16S rRNA gene sequencing, which produced a total of 20,772 Amplicon Sequence Variants (ASVs). (see revised pages 9,lines151-158)

- Please ensure that the newest taxa names are being used, see <https://www.nature.com/articles/s41522-024-00494-9>

Author response:

Thank you sincerely for offering me an opportunity to acquire

cutting-edge knowledge. This article is extremely helpful to me and I am highly inspired. While studying this article, I also referred to the contents of the 42 phyla of prokaryotes identified by the 2021 ICSP. The paper, published in Nature, expands on this by listing the names of 49 phyla that have been validly published.

After further verification, *Bacteroidota*, *Fusobacteriota*, and *Actinomyces* in the oral saliva samples in our manuscript were consistent with the validly published name. In the BALF samples, *Bacteroidota* and *Fusobacteriota* at the phylum level and *Pseudomonas* at the genus level correspond to the validly published names. For details, please see the Tables 1 and 2. The remaining phyla and genus were not found among the published names, but we ensure that these names are standard and widely used in the academic community.

- "Line 295 comment", the response was focused on not being able to rerun the data. However, please clarify what samples (OS or BALF or both combined) these are?

Author response:

Thank you very much for raising this question. Due to my negligence, I did not elaborate it clearly in our manuscript. Your reminder made me find this problem, and I have revised it in our manuscript. (see revised pages 22, lines 297-299)

It should be noted that, as shown in Figure 7, A, B and C represent the low, medium and high risk groups of PN respectively. The vertical coordinate represents the 30 most common bacterial genera obtained from the BALF and OS samples of each group of patients with PN, and the horizontal coordinate represents the 13 different bacterial genera obtained from the BALF and OS samples of each group of patients with PN.

The above are all our responses to the comments of you. Once again, thank you for your constructive suggestions. We believe that these modifications will significantly enhance the quality of our manuscript.

Re: Spectrum01284-24R2 (16S rRNA Sequencing Reveals Relationships among Enrichment of oral microbiota in the lower respiratory tract and Pulmonary Nodules Malignant Progression.)

Dear Dr. Xi Fu:

Your manuscript has been accepted, and I am forwarding it to the ASM production staff for publication. Your paper will first be checked to make sure all elements meet the technical requirements. ASM staff will contact you if anything needs to be revised before copyediting and production can begin. Otherwise, you will be notified when your proofs are ready to be viewed.

Sincerely,
Melissa Gitman
Editor
Microbiology Spectrum